# Understanding Alignment in Multimodal LLMs: A Comprehensive Study

## Abstract

Preference alignment has become a crucial component in enhancing the performance of Large Language Models (LLMs), yet its impact in Multimodal Large Language Models (MLLMs) remains comparatively underexplored. Similar to language models, MLLMs for image understanding tasks encounter challenges like hallucination. In MLLMs, hallucination can occur not only by stating incorrect facts but also by producing responses that are inconsistent with the image content. A primary objective of alignment for MLLMs is to encourage these models to align responses more closely with image information. Recently, multiple works have introduced preference datasets for MLLMs and examined different alignment methods, including Direct Preference Optimization (DPO) and Proximal Policy Optimization (PPO). However, due to variations in datasets, base model types, and alignment methods, it remains unclear which specific elements contribute most significantly to the reported improvements in these works. In this paper, we independently analyze each aspect of preference alignment in MLLMs. We start by categorizing the alignment algorithms into two groups, offline (such as DPO), and online (such as online-DPO), and show that combining offline and online methods can improve the performance of the model in certain scenarios. We review a variety of published multimodal preference datasets and discuss how the details of their construction impact model performance. Based on these insights, we introduce a novel way of creating multimodal preference data called Bias-Driven Hallucination Sampling (BDHS) that needs neither additional annotation nor external models, and show that it can achieve competitive performance to previously published alignment work for multimodal models across a range of benchmarks.

## 1 Introduction

Recent advancements in Multimodal Large Language Models (MLLMs) have significantly improved our understanding of vision-language tasks. By integrating visual signals with Large Language Models (LLMs), these models have demonstrated enhanced capabilities in multimodal understanding, reasoning, and interaction (Liu et al., 2023c; 2024; Bai et al., 2023; McKinzie et al., 2024).

Typically, MLLMs are pre-trained on large image-text datasets to develop foundational multimodal knowledge and skills, then undergo post-training for conversational capabilities, instruction following, helpfulness, and safety. Despite rapid advancements in recent years, significant challenges persist.

A notable problem is the tendency of MLLMs to produce responses that are not factually grounded in the visual input, commonly referred to as hallucinations, leading to inaccuracies such as incorrect descriptions of non-existent visual elements (Liu et al., 2023a; Cui et al., 2023). This undermines the trustworthiness of MLLMs in many practical applications.

Preference alignment methods have proven effective in reducing hallucinations and generating responses more closely aligned with human preferences for LLMs (Zhao et al., 2023b; Rafailov et al., 2023; Azar et al., 2024; Guo et al., 2024; Yuan et al., 2024; Ahmadian et al., 2024; Tang et al., 2024a). These methods utilize pairwise preference data to fine-tune the model, which can be based on Reinforcement Learning from Human Feedback (RLHF) (Christiano et al., 2017; Stiennon et al., 2020), Direct Alignment from Preferences (DAP) (Rafailov et al., 2023; Azar et al., 2024; Zhao et al., 2023b), or Online Direct Alignment from Preferences (Online-DAP) (Yuan et al., 2024; Guo et al., 2024).

While alignment in LLMs has been extensively studied, alignment for MLLMs has not yet been investigated to the same extent. Sun et al. (2023) and Zhou et al. (2024) aligned LLaVA 1.5 (Liu et al., 2023b) using Proximal Policy Optimization (PPO) and Direct Preference Optimization (DPO), respectively, while Li et al. (2023a) and Yu et al. (2023b) employed DPO and its variations to align Qwen-VL (Bai et al., 2023) and Muffin (Yu et al., 2023a) models. Notably, besides different alignment strategies and often different base models, all these works also introduce novel preference datasets for alignment with various sizes, collection, and generation schemes. As a result, while each of these studies offers valuable insights into alignment for MLLMs, it can sometimes be difficult to strongly attribute reported improvements to the individual proposed choices.

In this paper, we examine each component of multimodal alignment independently. First, we categorize alignment methods into two types: offline methods, which utilize preference pairs collected prior to training (e.g., DPO), and online methods, which involve sampling from the model during policy optimization (e.g., RLHF and Online-DAP). We conduct a comprehensive study over popular online and offline alignment methods, all aligning the popular LLaVA 1.6 model (Liu et al., 2024) using a fixed data regiment and study their benefits and shortcomings. To our knowledge, this is the first time that such study is conducted with MLLMs.

Further, we study the different methods for building pairwise preferences using public datasets. We break down the main components of preference data into three parts: prompts, chosen responses and rejected responses (Table 1). For each of those components, we investigate how their source, diversity, and quality can affect the resulting alignment. Additionally, we examine how the size of the alignment dataset impacts downstream performance.

Based on our comprehensive ablations, we identify a few key desiderata in alignment strategies for MLLMs and introduce a simple, novel preference data sampling scheme we call Bias-Driven Hallucination Sampling (BDHS). Despite not utilizing any human annotation nor the input of any external teacher model such as GPT4-V, we show that BDHS can achieve competitive performance against even much larger preference datasets constructed under different regimes.

## 2 ALIGNMENT

Preference alignment uses pairwise preference data. Each pair is linked to a text prompt, denoted as $x_{\text{text}}$, and an associated image, $x_{\text{img}}$, together forming the input $x = (x_{\text{img}}, x_{\text{text}})$. The responses include a preferred one, $y^+$, and a non-preferred or rejected one, $y^-$. See Section 3 for a more thorough discussion of these components. In this section, we focus on the various ways that a preference dataset, $\mathcal{D} = \{(x, y^+, y^-)\}_{i=1}^N$, is used by alignment approaches.

### 2.1 REINFORCEMENT LEARNING FROM HUMAN FEEDBACK (RLHF)

RLHF was the initial method used for alignment (Christiano et al., 2017; Stiennon et al., 2020), involving the training of a reward model (RM) from pairwise preferences and then optimizing a policy using the RM via reinforcement learning (RL). In RLHF, a reward model is initially trained on the preference pairs as described in Stiennon et al. (2020). The training of this reward model uses a straightforward cross-entropy loss, treating the binary choice – preferred or rejected – as a categorical label. The objective function for training the reward model, $r_\phi$, is as follows:

$$L_{\text{RM}} = -\log\left(\sigma\left(r_\phi(x, y^+) - r_\phi(x, y^-)\right)\right), \tag{1}$$

where $\sigma$ is the logistic function.

Next, the model (i.e., policy), $\pi_\theta$, is fine-tuned through RL using the trained reward model to optimize the following objective:

$$\max_{\pi_\theta} \mathbb{E}_{x \sim D, y \sim \pi_\theta(y|x)}\left[r_\phi(x, y) - \beta D_{\text{KL}}(\pi_\theta(y|x)|\pi_{\text{ref}}(y|x))\right]. \tag{2}$$

An additional KL penalty term $D_{\text{KL}}(\cdot)$ is incorporated to discourage significant deviations of $\pi_\theta$ from the initial model, $\pi_{\text{ref}}$ (Stiennon et al., 2020), and $\beta$ is a hyperparameter which adjusts the effect of this term in the overall objective.

Since the RM is learned in all RL-based approaches, it remains an imperfect approximation even when trained on human preferences. Previous work has shown that if not handled carefully, over-optimizing

for the RM can hurt the performance of the aligned model (Gao et al., 2023a). This adds a layer of challenge to these methods.

Different RL algorithms apply unique strategies to optimize the RL objective (Equation 2). In Appendix C we investigate the complexities of RL-based alignment for MLLMs, examining how different algorithms affect model performance. Specifically, we evaluate the impact of using PPO (Schulman et al., 2017; Stiennon et al., 2020; Ouyang et al., 2022) and REINFORCE Leave-One-Out (RLOO) (Williams, 1992; Ahmadian et al., 2024) in comparison to other alignment methods.

## 2.2 DIRECT ALIGNMENT FROM PREFERENCE

This family of approaches directly utilizes preference data, $D$, to optimize the policy, $\pi_\theta$. By eliminating the need to train a reward model, these methods significantly simplify the preference optimization pipeline. Furthermore, the gradient of all objectives can be precisely computed, distinguishing these methods from traditional RLHF approaches. The most widely used objective in MLLM alignment is DPO (Rafailov et al., 2023) (Equation 3). We have conducted the majority of our experiments using DPO to ensure comparability with other studies in MLLM alignment. In Appendix E, we also examine DPO alongside two other common offline methods, IPO (Azar et al., 2024) and SLiC (Zhao et al., 2023b). For a unified derivation of common direct alignment methods refer to Tang et al. (2024b). For brevity, we only recap the DPO loss function:

$$L_{\text{DPO}}(\pi_\theta; \pi_{\text{ref}}) = \mathbb{E}_{(x,y^+,y^-)\sim D} \left[ -\log \sigma \left( \beta \log \frac{\pi_\theta(y^+|x)\pi_{\text{ref}}(y^-|x)}{\pi_{\text{ref}}(y^+|x)\pi_\theta(y^-|x)} \right) \right] . \tag{3}$$

We will omit the dependency on $\pi_{\text{ref}}$ from subsequent equations for simplicity.

It is important to note that most preference datasets are not derived from the model being aligned and are collected offline. Even when the data is constructed based on the model that is undergoing alignment, the samples encountered during training do not account for changes in the model over training. This leads to a distribution shift between the model that generated the data and the model being aligned, which can be considered a disadvantage of these methods.

## 2.3 ONLINE DIRECT ALIGNMENT FROM PREFERENCE

Recently, a new family of algorithms has been proposed for aligning LLMs. These methods do not train a separate reward model. Instead, they employ either the model that is being aligned (Yuan et al., 2024) or a different LLM (Guo et al., 2024) to obtain online feedback to create preference pairs. These pairs are then used to optimize the objective function via for example DPO. This approach eliminates the complexity of training a separate reward model while still taking advantage of online samples from the model, thereby avoiding distribution shifts.

We explore the use of LLaVA 1.6-34B (Liu et al., 2024) as annotator to generate online preference pairs, motivated by its strong performance on a multitude of multimodal benchmarks. Additionally, we investigate a hybrid approach that combines online and offline approaches. This method involves sampling from the offline preference data with a probability $p$, $(y^+, y^-)$, and sampling from the model with a probability $1 - p$, $(\tilde{y}^+, \tilde{y}^-)$. Equation 4 details this approach.

$$L_{\text{Mixed-DPO}}(\pi_\theta) = \mathbb{E}_{\substack{(x,y^+,y^-)\sim D \\ (\tilde{y}^+,\tilde{y}^-)\sim\pi_\theta}} \left[ \alpha L_{\text{DPO}}(y^+, y^-, x; \pi_\theta) + (1-\alpha)L_{\text{DPO}}(\tilde{y}^+, \tilde{y}^-, x; \pi_\theta) \right] , \tag{4}$$

where $\alpha \sim \text{Bernoulli}(p)$. In our experiments we use $p = 0.5$. This algorithm is similar to techniques used in off-policy RL methods like Q-learning (Hester et al., 2018), where a replay buffer includes samples from both the model and expert demonstrations. We found this approach particularly effective when the online and offline methods have complementary effects on the model's final performance.

## 3 MULTIMODAL PREFERENCE DATA

Preference datasets have three main components: the prompts, the chosen and the rejected responses. Responses are usually constructed by prompting one or more MLLMs, typically excluding the model being aligned. Those responses are associated with a preference signal. There are two

| Type | Name | Size | Prompt | | Response | | Judge | Preference |
|---|---|---|---|---|---|---|---|---|
| | | | Text | Image | Chosen | Rejected | | Signal |
| Human | LLaVA-RLHF | 10k | LLaVA-Instruct-150k | COCO | LLaVA 1.5 | LLaVA 1.5 | human | ranking |
| | RLHF-V | 5.7k† | UniMM-Chat | Various† | Muffin/Various†(corrected) | Muffin/Various † | human | construction |
| Synthetic | VLFeedback | 80k | 9 datasets (LLaVA, SVIT, etc.) | | 12 MLLMs (LLaVA 1.5, GPT-4V, etc.) | | GPT-4V | ranking |
| | NLF | 63k | LLaVA-Instruct-150k | COCO | DRESS_ft (refined) | DRESS_ft | GPT-4V | construction |
| | POVID | 17k | LLaVA-Instruct-150k | COCO | SFT Ground truth | SFT Ground truth (corrupted) | GPT-4V | construction |

Table 1: Recently published multimodal preference datasets. † denotes the updated dataset version.

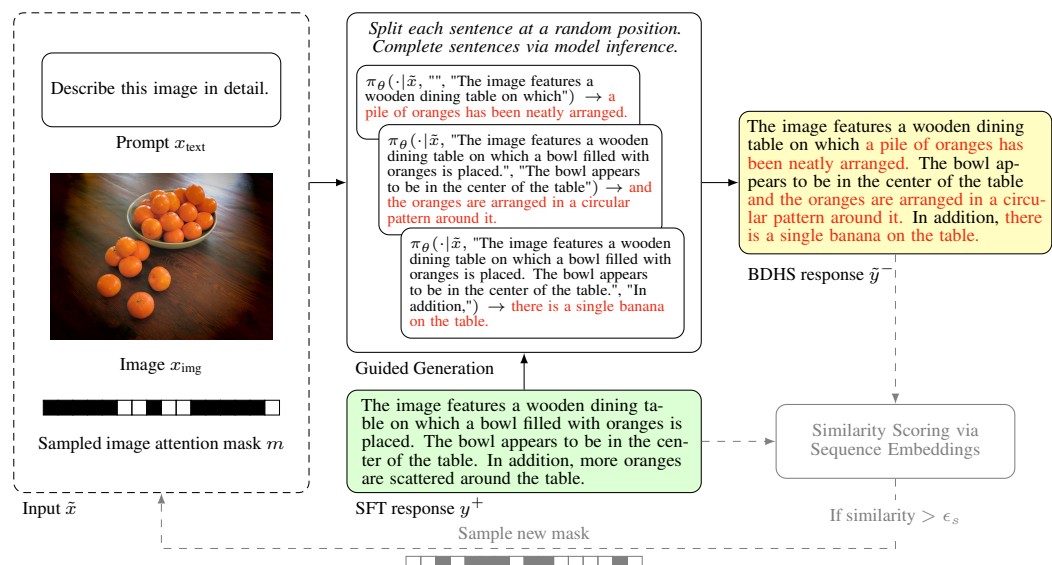

Figure 1: Overview of the BDHS method including the optional iterative variant in gray. For each re-generation, both the image attention mask and the sentence split positions are resampled. The image is taken from the LLaVA Instruct dataset.

sources: (a) datasets that rely on **human annotators** to compose the preference, such as LLaVA-RLHF (Sun et al., 2023) or RLHF-V Yu et al. (2023b), and (b) datasets with **synthetic annotations**, commonly originating from GPT-4V acting as a ranker or corrupter, such as DRESS (Chen et al., 2023), VLFeedback (Li et al., 2023a) and POVID Zhou et al. (2024). Table 1 presents those datasets.

Another way to organize those datasets is to consider the nature of the preference signal. LLaVA-RLHF and VLFeedback use **ranking**: responses are sampled, then ranked by humans or GPT-4V. Other works use a **construction** approach: signal is obtained by correcting responses, such as RLHF-V and DRESS, or by corrupting them such as POVID. In Appendix F.1, we provide more details.

One constant, however, among those methods is the cost to build the preference – whether using a strong MLLM or humans. In the next section, we propose a method to mitigate this requirement.

### 3.1 BIAS-DRIVEN HALLUCINATION SAMPLING (BDHS)

Hallucinations in MLLMs often express the underlying language models' inherent biases, for example towards frequently cooccuring objects or object attributes (Li et al., 2023b; Qian et al., 2024; Zhou et al., 2024). In other words, the MLLM may choose to draw from its parametric knowledge or textual context $x_{text}$ when instead it should have more strongly considered information from the image $x_{img}$ in question. Zhou et al. (2024) suggest to trigger inherent biases directly by presenting noisy images $\tilde{x}_{img}$ to the model when generating the non-preferred response $\tilde{y}^-$ via teacher-forcing (POVID-style image distortion). While this method has desirable characteristics, such as not requiring external teacher models or human annotation to construct preference pairs, as well as generating samples that are at least partially informed by the policy under alignment, it carries some notable drawbacks. Zhou et al. (2024) show that selecting too few diffusion steps can yield insufficient corruption, whereas

too many diffusion steps negatively impacts the generated responses, as the model mainly identifies noise respective to *pixels*. In our experiments, we further found that the proposed teacher-forcing can introduce non-sensical responses. Further details about POVID-style image distortion are provided in Appendix G.1 and an example with non-sensical responses in Appendix G.7.

Inspired by the POVID work, we aim to address its main identified shortcomings: 1. We propose to rethink the method of corrupting the signal from the input image from a pixel-based approach to one that limits access in the latent space via attention masking, which we argue more directly achieves the underlying motivation of triggering the inherent bias of the underlying language model. 2. We introduce a new reference-guided generation strategy that allows corrupted responses to remain largely true to the chosen response while still introducing meaningful divergence, without introducing non-sensical continuations introduced by token-based teacher forcing. 3. We use an off-the-shelf sentence embedding to verify that the generated rejected response is meaningfully distinct from the original reference to focus the resulting feedback signal on hallucinations over mere stylistic difference. We refer to our novel technique as Bias-Driven Hallucination Sampling (BDHS). BDHS is annotation free and computationally efficient to the point that rejected responses can be generated online, which we explore in Section 4.4. An overview of the method can be found in Figure 1, with further details provided in the following subsections.

Let $\tilde{x} = (x_{\text{text}}, \tilde{x}_{\text{img}}, m)$ denote the modified input with (optional noisy) image $\tilde{x}_{\text{img}}$ and attention mask $m$. Suppose the MLLM encodes image $\tilde{x}_{\text{img}}$ to $k$ embedding vectors, each vector with dimension $d$. Then, $m$ is defined as a boolean mask of dimension $k$. We suggest to randomly sample the mask $m = (m_1, m_2, \ldots, m_k)$ according to a uniform distribution $\mathcal{U}(0, 1)$ and threshold $\rho_{\text{th}} \in [0, 1]$ where each element $m_i = 1$ if $\rho_i \geq \rho_{\text{th}}$ for $\rho_i \sim \mathcal{U}(0, 1)$ or $m_i = 0$ otherwise. By masking the image embeddings using $m$, the model only pays attention to a subset of the $k$ embedding vectors to generate the response $\tilde{y}^-$. Where the remaining signal is not sufficient, the MLLM can only draw on its parametric knowledge to answer, thus inducing hallucination. By allowing access to some part of the image, we encourage more realistic hallucinations.

Keeping the generated corrupted response $\tilde{y}^-$ close to the preferred one, $y^+$, supports the optimizer in paying more attention to the image as only the non-overlapping portion is affected by the modified input $\tilde{x}$. Otherwise, responses $\tilde{y}^-$ and $y^+$ could diverge early on or $\tilde{y}^-$ could even represent a generic response hinting on missing image information. Instead, in order to maintain consistency in style and structure we propose a reference-guided sampling strategy, where we "diverge" and "rejoin" from $y^+$ at random points in every sentence to form $\tilde{y}^-$. A formal description of the algorithm is provided in Appendix G.3.

Similar to our observation in Online-DAP, BDHS responses $\tilde{y}^-$ can still be very similar to the ground truth $y^+$, especially when the pivot position is late in the sentence. To maximize learning utility of BDHS preference pairs, this is undesirable.

While further increasing $\rho_{\text{th}}$ or biasing towards early pivot positions in the reference-guided generation could minimize such trivial generations, this introduces additional hyperparameters and can lead to less realistic dispreferred responses. Instead, we realize BDHS in an iterative fashion.

Once $\tilde{y}^-$ is generated, a semantic similarity score w.r.t. $y^+$ is computed using an off-the-shelf sentence embedding model[1]. A new response $\tilde{y}^-$ is sampled if the cosine similarity is above a pre-defined threshold $\epsilon_s$. After reaching the maximum number of iterations $N_{\text{BDHS}}$, $\tilde{y}^-$ is generated according to input $\tilde{x}$ without any reference guidance. Appendix G provides the actual algorithm for BDHS including similarity scoring and several examples.

This additional comparison avoids $\tilde{y}^-$ responses that are trivial rephrasings. Moreover, measuring the number of examples that need re-generation allows intuitive tuning of the $\rho_{\text{th}}$ hyper parameter.

## 4 EXPERIMENTS

In this section, we empirically evaluate different aspects of aligning MLLMs. We start by summarizing our key findings in Section 4.1. We conduct our ablations on the LLaVA 1.6-7B Vicuna model, as it is both well studied and exhibits relatively strong performance across a range of multimodal tasks (Liu et al., 2024). Notably, this model provides stronger baseline performance over the more

---

[1] We use the *all-mpnet-base-v2* sentence embedding model (Reimers & Gurevych, 2019) with $\epsilon_s = 0.97$.

| Model | Alignment | Dataset | POPE ↑ | MMHAL ↑ | MMHAL$^v$ ↑ | LLaVA$^W$ ↑ | VQA$^T$ ↑ | GQA ↑ | MMVet ↑ | Recall$^{coco}$ ↑ |
|---|---|---|---|---|---|---|---|---|---|---|
| LLaVA 1.6-7B | – | – | 86.40 | 2.95 | 2.75 | 80.85 | 64.85 | 64.23 | 43.94 | 68.13 |
| LLaVA 1.6-13B | – | – | 86.23 | **3.23** | **3.18** | 86.10 | **65.7** | **64.8** | 48.26 | 68.13 |
| LLaVA 1.6-34B | – | – | 87.73 | **3.50** | **3.46** | 88.35 | **69.5** | **67.1** | 53.90 | 71.17 |
| OmniLMM-12B† | – | – | – | 3.14 | – | 74.3 | – | – | – | – |
| LLaVA 1.6-7B† | DPO | STIC | – | – | – | 79.2 | 65.2 | – | 45.0 | – |
| LLaVA 1.5-7B† | RLAIF-V | RLAIF-V | – | 3.06 | – | 64.9 | – | – | – | – |
| OmniLMM-12B† | RLAIF-V | RLAIF-V | – | **3.36** | – | 74.3 | – | – | – | – |
| LLaVA 1.6-7B | DPO | POVID (Full) | 88.09 | 3.16 | 3.07 | 78.63 | 64.56 | 64.12 | 40.60 | 73.48 |
| LLaVA 1.6-7B | Online-DPO | POVID (Full) | 86.49 | 2.88 | 2.94 | 82.61 | 64.88 | 64.31 | 43.26 | 68.45 |
| LLaVA 1.6-7B | Mixed-DPO | POVID (Full) | 88.03 | 2.83 | 3.10 | 82.75 | 64.93 | **64.47** | 42.80 | 74.53 |
| LLaVA 1.6-7B | DPO | POVID (Full) | 88.09 | 3.16 | 3.07 | 78.63 | 64.56 | 64.12 | 40.60 | 73.48 |
| LLaVA 1.6-7B | DPO | BDHS (POVID, 5k) | 88.75 | 2.61 | 2.71 | 86.33 | 65.07 | 63.97 | 43.4 | **75.58** |
| LLaVA 1.6-7B | DPO | Online-BDHS (POVID, 5k) | **88.83** | 2.80 | 2.99 | 85.03 | 65.09 | 63.65 | 43.12 | 74.09 |
| LLaVA 1.6-7B | DPO | ∗ ∪ POVID (5k) | 88.38 | 2.82 | 2.81 | 84.01 | **65.42** | 64.30 | **45.46** | 74.00 |
| LLaVA 1.6-7B | DPO | VLFeedback (Full) | 81.84 | 2.96 | 2.99 | **90.75** | 62.93 | 62.53 | 43.85 | 66.67 |
| LLaVA 1.6-7B | DPO | VLFeedbackCorrupted (5k) | 87.52 | 3.03 | 3.01 | 88.64 | 65.30 | 64.19 | 42.16 | 70.13 |
| LLaVA 1.6-7B | DPO | BDHS (VLFeedback, 5k) | 88.10 | 2.77 | 2.87 | 86.68 | 65.27 | 64.33 | 43.39 | 72.43 |

Table 2: Main results. The best and second best results are shown in **bold** and underlined, respectively. If a larger model outperforms all aligned 7B models, it is indicated by **bold and underline**. † denotes results reported from referenced papers, and a dash (–) marks benchmarks that are not reported. Rows in blue are contributions of this paper.

common choice of LLaVA 1.5-7B in the multimodal alignment literature and we generally observe correspondingly smaller relative improvements from alignment. We carefully select a set of benchmarks to measure overall helpfulness and hallucination propensity. Our study reveals shortcomings in existing benchmarks, particularly around measuring hallucinations. Please refer to Appendix B.1 for details. We also include an in-depth ablation study on the components we have discussed in the paper, offering a clearer view of effect. We begin with equalizing the experimental conditions on public preference datasets (Section 4.2). We then highlight desiderata for a high-quality preference dataset (Section 4.3) and show that BDHS can be a simple and effective mechanism following such best practices (Section 4.4). Additionally, we compare RL-based methods (Appendix C), Online and Mixed-DPO strategies (Appendix D.3), as well as various offline approaches (Appendix E).

## 4.1 KEY COMPONENTS IN MLLM ALIGNMENT PIPELINE

We summarize our main findings and compare results with other SOTA models in Table 2. First, we fixed the base model (LLaVA 1.6-7B) and studied the effects of online vs. offline methods using the POVID alignment data (Zhou et al., 2024). While offline DPO shows more significant improvement on benchmarks that consider hallucination, such as POPE and MMHALBench-V, Online-DPO enhances benchmarks evaluating the quality of answers in an open question answering setup, like LLaVABench-in-the-Wild. This is intuitive, as the preference pairs in POVID are specifically designed to reduce hallucinations whereas the online samples from the model may provide other signals. Mixed-DPO allows to incorporate the benefits of both approaches and the results show consistent improvement over both online and offline methods.

When using Online-DPO or Mixed-DPO strategies, we typically depend on advanced models like LLaVA 1.6-34B to rank the online samples generated by the model. However, access to such models is not always guaranteed. We discuss this limitation in more detail in Appendix D.2. Additionally, the construction of the POVID dataset also involves using a superior model such as GPT-4V to inject noise into SFT data. Our proposed BDHS method does not require additional annotators or preference data, and relies exclusively on SFT data already available from the instruction tuning of the base model. Despite this simplicity, it consistently outperforms the models that utilize the larger POVID dataset (i.e. both offline and Mixed-DPO) in most benchmarks. Implementing BDHS in an online format further closes this performance gap in MMHALBench-V, establishing BDHS as a compelling and cost-effective alternative to other more resource-intensive approaches. Combining the POVID dataset with the online-BDHS approach (referred to as Online-BDHS ∪ POVID), with the exception of MMHALBench-V, consistently outperforms the model that uses only the POVID dataset across all benchmarks. It also surpasses STIC (Deng et al., 2024) and RLAIF-V (Yu et al., 2024) on the reported benchmarks. We further discuss the enhanced efficacy of our approach over Zhou et al. (2024) in Section 4.4.

While Section 4.3 provides a detailed analysis of various preference datasets, we highlight key findings from the VLFeedback dataset here, as they contribute significantly to building an effective alignment strategy. Unlike POVID, both VLFeedback and its variant, VLFeedbackCorrupted(5k),

| Model | Dataset | POPE ↑ | MMHAL ↑ | MMHAL$^V$ ↑ | LLaVA$^W$ ↑ | VQA$^T$ ↑ | GQA ↑ | MMVet ↑ | Recall$^{coco}$ ↑ |
|---|---|---|---|---|---|---|---|---|---|
| LLaVA 1.6-7B | – | 86.40 | 2.95 | 2.75 | 80.85 | 64.85 | 64.23 | 43.94 | 68.13 |
| **Public datasets** | | | | | | | | | |
| LLaVA 1.6-7B | VLFeedback (80k) | 81.84 | 2.96 | 2.99 | 90.55 | 62.93 | 62.54 | 43.85 | 66.67 |
| LLaVA 1.6-7B | POVID (17k) | 88.09 | 3.16 | 3.07 | 78.05 | 64.56 | 64.12 | 40.60 | 73.48 |
| LLaVA 1.6-7B | RLHF-V (5.7k) | 83.86 | 3.15 | 3.26 | 70.58 | 64.75 | 62.89 | 37.16 | 64.26 |
| **Public datasets, randomly subsampled to 5,000 samples** | | | | | | | | | |
| LLaVA 1.6-7B | VLFeedback (5k) | 86.31 | 2.92 | 3.00 | 83.10 | 65.06 | 64.09 | 43.21 | 68.03 |
| LLaVA 1.6-7B | POVID (5k) | 88.18 | 2.93 | 2.93 | 81.89 | 64.90 | 64.34 | 43.39 | 71.80 |
| LLaVA 1.6-7B | RLHF-V (5k) | 84.39 | 3.25 | 3.35 | 72.09 | 64.85 | 63.35 | 39.72 | 64.68 |
| **Previously published** | | | | | | | | | |
| Qwen-VL-Chat | VLFeedback (Li et al., 2023a) | – | 3.02 | – | – | – | – | 49.9 | – |
| Muffin | RLHF-V (Yu et al., 2023b) | – | (52.1↓)† | – | – | – | – | – | – |
| LLaVA 1.5 | POVID (Zhou et al., 2024) | 86.90 | 2.69 | – | 68.7 | – | – | 31.8 | – |

Table 3: Results for LLaVA 1.6-7B Vicuna (Liu et al., 2024) aligned with DPO on VLFeedback, POVID, RLHF-V. Results highlighted in gray are the results reported by the original authors. † denotes `MMHALBench` for which Yu et al. (2023b) strictly reported the human-corrected hallucination rate.

select the "chosen response" in the preference pairs from the top responses ranked by GPT-4V, selected from a pool of model-generated responses. Compared to re-using SFT data, this approach potentially offers an additional supervisory signal to the model, leading to enhanced performance on benchmarks like `LLaVABench-in-the-Wild`, where such aligned models even outperform the unaligned 13B and 34B models from the same family.

Notably, we introduce VLFeedbackCorrupted (5k), a small dataset leveraging corruption injection to generate the "rejected response", which performs competitively to the much larger ranking-based VLFeedback (full) dataset. These experiments demonstrate the effectiveness of two strategies in constructing preference data: First, learning from strong (highly-ranked) responses seems to yield a distillation-like benefit. Second, using subtle differences between "chosen" and "rejected" responses, as opposed to just rank-based pairs (like in VLFeedback (full)), can significantly reduce hallucinations, even in a limited data regiment.

Finally, we replace the noise injection strategy using GPT-4 with our proposed BDHS. We observe a slight reduction of the `MMHALBench-V` and `LLaVABench-in-the-Wild` scores compared to the GPT-4V based approach, but note that the achieved result still represents meaningful improvements over the baseline. On all other metrics, BDHS shows comparable or even superior results, establishing BDHS as a strong alternative to GPT-4V in this pipeline.

In the rest of this section, we conduct a comprehensive ablation study on each of the components discussed earlier, aiming to offer insights into the typical trade-offs encountered in alignment strategies.

## 4.2 REMOVING CONFOUNDING FACTORS FOR PREVIOUSLY PUBLISHED DATASETS

We analyze RLHF-V (Yu et al., 2023b), VLFeedback (Li et al., 2023a) and POVID (Zhou et al., 2024) as they offer a fair blend between human and synthetic sources, and between constructed and ranked preference signal composition. As it is challenging to determine what are the properties that characterize a high-quality preference dataset, we first replicate alignment using the published datasets against LLaVA 1.6-7B with DPO. Additionally, we sub-sample all datasets to a consistent size of 5,000 examples to remove effect sizes. Results are reported in Table 3.

Zhou et al. (2024) have conducted a similar experiment using LLaVA 1.5, however they do not control for dataset size. We were successful in replicating certain observations published by these authors. POVID reaches the highest score on `POPE`. Zhou et al. (2024) also reports the highest `MMHALBench` scores with POVID, which we were able to reproduce using LLaVA 1.6, although this is only true when size correction is not applied. Upon normalizing for size, POVID's performance equaled that of VLFeedback and was lower than RLHF-V.

In other domains, our experiment have shown divergent trends. While Zhou et al. (2024) demonstrated that all preference datasets improved LLaVA 1.5 on `MMVet`, our findings with LLaVA 1.6 exhibited a reverse trend: all our runs did not match up to the baseline. Interestingly, as the datasets grew larger, we witnessed a further deviation from the baseline. We hypothesize that these preference datasets lack the necessary information to improve `MMVet` over the notably stronger baseline LLaVA 1.6

introduced, which necessitates specialized knowledge (see Appendix B.1). VLFeedback, to a certain extent, may possess some of this knowledge thanks to its diverse prompts. By restricting dataset sizes, we further limit the potential alterations on the non-aligned model, as the results stay closer to the baseline.

Notable, VLFeedback moderately improves `LLaVABench-in-the-Wild` when the size restriction limit is applied. When aligning on the complete VLFeedback, the largest dataset in these experiments, we see further improvement and can achieve the highest score on that benchmark.

### 4.3 DESIDERATA FOR PREFERENCE DATASETS

We examine the components of a multimodal preference dataset and investigate the following options. The explored choices are further summarized in Table 4.

- **Prompts** We compared (i) a diverse prompt strategy mixing multiple datasets to (ii) prompts only from LLaVA-Instruct-150k, which was already seen during the SFT stage of the base model.
- **Chosen responses** We introduced 3 settings: (i) diverse responses from multiple MLLMs; (ii) LLaVA responses only, (iii) GPT-4V responses only.
- **Rejected responses** We introduced 2 settings: (i) diverse responses from multiple MLLMs, and (ii) corruption of the chosen responses.

In order to construct these preference dataset ablations cheaply and reproducibly, we leverage the size and diversity of the VLFeedback dataset (Li et al., 2023a). VLFeedback possesses several properties that makes it a good sandbox: (a) the prompts, derived from 9 datasets (LLaVA-Instruct-150k, SVIT, LLaVAR, etc.), are diverse, (b) the chosen and rejected responses are sampled from 12 MLLMs making them very diverse too – $\sim 37\%$ responses are from GPT-4V, and $\sim 35\%$ from the LLaVA 1.5 series, (c) finally, the large size of VLFeedback, 80,000 quadruplets of responses that can be paired together, makes it simpler to isolate specific aspects.

| Datasets | Prompts | | Chosen Responses | | | Rejected Responses | |
|---|---|---|---|---|---|---|---|
| | diverse | LLaVA-SFT | diverse | LLaVA | GPT-4V | diverse | chosen corrupted by GPT-4 |
| VLFeedback | ✓ | | ✓ | | | ✓ | |
| + corrupting strategy | ✓ | | ✓ | | | | ✓ |
| **prompts** | | | | | | | |
| LLaVA prompts | | ✓ | ✓ | | | | ✓ |
| **model responses** | | | | | | | |
| GPT-4V responses only | | ✓ | | | ✓ | | ✓ |
| LLaVA responses only | | ✓ | | ✓ | | | ✓ |

Table 4: Controlled settings for multimodal preference dataset exploration. We decompose the preference datasets into prompts, chosen and rejected responses and we then aim at identifying factors contributing to the dataset quality.

**Corruption strategy** Reranking is originally used to determine chosen and rejected responses in VLFeedback (see Section 3). In order to remove variation introduced by the original rejected responses (e.g., style change between MLLMs) and permit a tighter control on ablations, we replace rejected responses from the original VLFeedback samples with corrupted versions of the preferred responses. Similar to the method in (Zhou et al., 2024), we leverage GPT-4 to specifically introduce realistic hallucinations, assisted by a few shots for illustration (see Appendix F.2).

**Results** Following Section 4.2, we apply DPO alignment on the LLaVA 1.6-7B model, and we limit all the datasets to 5,000 samples. In Table 5, we report the results of this experiment. First, we show that our corruption strategy achieves improvements over the baseline comparable in magnitude to the ranking-based preference signal in the original VLFeedback data. In some benchmarks, like `MMHAL-Bench-V`, we even observe improvements, while notably `MMVet` shows some regressions. Nevertheless, we argue that this represents a reasonable baseline to adopt for easier iteration on the following ablations. In Appendix F.3, we provide more analysis on this strategy.

Next, we explore the impact of novelty of the prompts used for alignment, by sampling another 5k preference data generated with the same corruption mechanism solely from prompts that are a part of the LLaVA SFT mixture. These are examples that the baseline model would have already been trained

| Dataset | POPE ↑ | MMHAL ↑ | MMHAL$^v$ ↑ | LLaVA$^W$ ↑ | VQA$^T$ ↑ | GQA ↑ | MMVet ↑ | Recall$^{coco}$ ↑ |
|---|---|---|---|---|---|---|---|---|
| Baseline | 86.40 | 2.95 | 2.75 | 80.85 | 64.85 | 64.23 | 43.94 | 68.13 |
| VLFeedback (5k) | 86.31 | 2.92 | 3.00 | 83.10 | 65.06 | 64.09 | 43.21 | 68.03 |
| + corrupting strategy | 85.59 | 3.39 | 3.33 | 86.65 | 65.20 | 63.87 | 37.98 | 68.66 |
| **prompts** | | | | | | | | |
| LLaVA prompts | 87.63 | 2.85 | 2.96 | 86.55 | 65.13 | 64.25 | 41.47 | 70.44 |
| **model responses** | | | | | | | | |
| GPT-4V responses only | 86.78 | 3.30 | 3.02 | 86.77 | 65.06 | 64.02 | 40.14 | 69.08 |
| LLaVA responses only | 87.52 | 3.03 | 3.01 | 88.64 | 65.30 | 64.19 | 42.16 | 70.13 |

Table 5: We started from VLFeedback with its diverse prompts and responses, and we then applied targeted sampling and corruption to isolate factors contributing to the dataset quality.

on during the SFT stage, so one may argue the baseline has already been taught a desirable response for. Interestingly, we observe that this shows similar improvement. We still observe comparable lift on `LLaVABench-in-the-Wild`, and while `MMHAL-Bench-V` shows less dramatic improvement over the baseline compared to the more diverse corruption-based sample, this may be due to more verbose responses, as indicated by higher recall. `POPE` even improves somewhat significantly and the regression in `MMVet` is also less pronounced.

Finally, we explore the impact of the construction of the accepted response in the alignment data. One could argue that for responses derived from stronger model such as GPT-4V, improvements may also be the result of learning from this stronger teacher model. Therefore, we conduct two experiments: one, where we sample data where the preferred response comes from GPT-4V only, and one where the preferred response comes from LLaVA 1.5-7B, a model generally weaker than the base model under alignment in this experiment. Interestingly, we do not observe any benefit from learning from GPT-4V generated responses, in fact, our results suggest that positive samples derived from LLaVA 1.5-7B led to a slightly stronger model post alignment.

These findings suggests that useful preference data can be derived cheaply, even from responses from relatively weaker models, as long as one can effectively sample and identify relatively desirable answers from the model as their preferred response, and introduce targeted corruption to create dispeferred responses.

## 4.4 ABLATIONS ON BDHS

Section 3.1 introduces BDHS as a technique to generate corrupted responses directly using the model subject to alignment. While our proposed approach is purely based on image attention masking, we also evaluate a variant that consumes noisy images instead, motivated by the teacher-forced *POVID-style image distortion* introduced in Zhou et al. (2024) (see Section G.1). In the following, BDHS with attention masking ($\rho_{th} = 0.99$ and $N_{BDHS} = 5$) is denoted as BDHS$_{attn}$ and BDHS with noisy images in the input as BDHS$_{noise}$. The number of additive noise steps for BDHS$_{noise}$ is set to $N = 500$ similar to the image distortion in Zhou et al. (2024).

All ablations in Table 6 are based on our 5k subset of POVID as introduced in Section 4.2. POVID contains LLaVA Instruct responses $y^+$ as well as GPT-4V corrupted non-preferred responses $y^-$. While $y^+$ is shared between all ablations, we start with substituting $y^-$ from external supervision by the BDHS model response $\tilde{y}^-$ and invoke standard DPO as shown in the first 3 rows after the LLaVA 1.6-7B baseline results. The proposed variants consistently improve over the baseline for `POPE` and `LLaVABench-in-the-Wild`. They regress on `MMHALBench`, however, as discussed in Section B.1, this benchmark has limitations so we mainly focus on `MMHALBench-V` instead for which all BDHS$_{attn}$ variants perform comparable to the baseline while the online rollout of $\tilde{y}^-$ even improves over it. Notably, we also observe significantly higher `Recall`$^{coco}$, suggesting richer responses. BDHS$_{noise}$ results in lower scores for `LLaVABench-in-the-Wild` while the attention masking approach BDHS$_{attn}$ almost maintains the baseline scores.

The lower partition of Table 6 starts with plain DPO on the POVID (5k) dataset as reference and then each subsequent approach incorporates both the existing response $y^-$ from external supervision as well as $\tilde{y}^-$ derived from the policy. Hereby, the two non-preferred responses are incorporated into the DAP framework by averaging the losses of $(y^+, y^-)$ and $(y^+, \tilde{y}^-)$ according to Equation equation 5.

| $y^-$ from external supervision | $\tilde{y}^-$ derived from policy | POPE↑ | MMHAL↑ | MMHAL$^V$ ↑ | LLaVA$^W$↑ | VQA$^T$ ↑ | GQA↑ | MMVet↑ | Recall$^{coco}$ ↑ |
|---|---|---|---|---|---|---|---|---|---|
| – | – | 86.40 | **2.95** | 2.75 | 80.85 | 64.85 | 64.23 | 43.94 | 68.13 |
| – | BDHS$_{noise}$ (Offline, ours) | 88.60 | 2.37 | 2.48 | 84.53 | 65.05 | 64.14 | 41.38 | 75.16 |
| – | BDHS$_{attn}$ (Offline, ours) | 88.75 | 2.61 | 2.71 | **86.33** | 65.07 | 63.97 | 43.4 | **75.58** |
| – | BDHS$_{attn}$ (Online, ours) | **88.83** | 2.80 | **2.99** | 85.03 | 65.09 | 63.65 | 43.12 | 74.09 |
| GPT-4V (POVID) | – | 88.18 | 2.93 | 2.93 | 81.89 | 64.90 | 64.34 | 43.39 | 71.80 |
| GPT-4V (POVID) | POVID-style image distortion | 88.33 | 2.84 | 2.64 | 80.15 | 64.21 | 63.79 | 41.28 | 69.39 |
| GPT-4V (POVID) | BDHS$_{noise}$ (Offline, ours) | 88.58 | 2.76 | 2.45 | 84.36 | 65.31 | 64.26 | 43.95 | 75.05 |
| GPT-4V (POVID) | BDHS$_{attn}$ (Offline, ours) | 88.56 | 2.85 | 2.72 | 85.35 | 65.39 | 64.11 | 43.26 | 75.05 |
| GPT-4V (POVID) | BDHS$_{attn}$ (Online, ours) | 88.38 | 2.82 | 2.81 | 84.01 | **65.42** | 64.30 | **45.46** | 74.00 |

Table 6: Ablation results for BDHS including baseline and reference approaches. All results based on LLaVA 1.6-7B, using DPO and the POVID (5k) sample for the source of images and prompt. Whenever both $y^-$ from external supervision and $\tilde{y}^-$ derived from policy (either online or offline) are incorporated, the average loss is computed using equation 5.

Therefore, the Online-BDHS method uses Online-DPO in a considerable simplified setting compared to the full Online-DPO realization equation 4, as the formulation presented here does not depend on a dedicated external annotator (see Section 2.3).

All the BDHS ablations improve significantly on `LLaVABench-in-the-Wild` compared to the DPO baseline and POVID-style image distortion. The BDHS$_{attn}$ with attention masking performs significantly better on `MMVet` compared to BDHS$_{noise}$. Notably, BDHS$_{attn}$ consistently outperforms the POVID-style image distortion across all benchmarks. We follow the published implementation of Zhou et al. (2024), however, surprisingly the *POVID-style image distortion* performs worse compared to plain DPO via POVID (5k), which differs from the LLaVA 1.5-7B alignment results in their paper. Presumably, the non-sensical responses from teacher-forcing could lower the performance while trading off with the existing GPT4-V preference pairs.

While online approaches with BDHS improve on certain benchmarks, we emphasize that even the offline dataset created with BDHS$_{attn}$ and without additional response from external supervision already constitutes a cost-effective baseline that consistently performs well across all benchmarks.

# 5 CONCLUSION AND FUTURE WORK

In this study, we investigate preference alignment's role in improving MLLM performance, particularly in reducing hallucinations. We evaluate various alignment strategies, including offline, online, and hybrid approaches, across different datasets. Our analysis of existing multimodal preference datasets allows for directly comparing their impact on model performance, controlled for data size and the base model under alignment. We further isolate individual differences between these datasets via additional ablations and show that corruption-based preference data can achieve hallucination reduction comparable to sampling and ranking-based approaches, and that learning from novel and diverse inputs, or from responses from superior models surprisingly does not lead to further improvements. Based on these insights, we present a simple preference dataset generation strategy we call BDHS, which uses only existing SFT data, eliminating the need for a superior model, human labelers, or other complex means of constructing preference data. Applied to LLaVA 1.6, this simple methods yield significant improvements across benchmarks, demonstrating its potential.

This study not only enhances our understanding of preference alignment but also establishes a foundation for further research into MLLM preference alignment. We identify several gaps in the community's approach to aligning MLLMs. Firstly, while considerable research has been conducted on various alignment methods for LLMs, including both online and offline approaches, these studies are less common in the context of MLLMs. For instance, RLH(AI)F is extensively discussed in LLM literature, highlighting its potential over simpler methods like DPO. Although we provide some insights into RL-based alignment for MLLMs and the evaluation of reward models, a significant gap remains between LLM and MLLM research in this domain. Secondly, we emphasize the need for better hallucination benchmarks to help understand model improvements. Moreover, our thorough analysis of published multimodal preference data reveals insufficient coverage in existing datasets, which may contribute to the absence of significant improvements in some benchmarks. Lastly, while BDHS effectively generates preference data to reduce hallucination, further research is needed to address other factors causing hallucination in MLLMs, such as insufficient real-world knowledge.

We hope these insights inspires further research and helps the community tackle ongoing challenges in this field.

## 6 REPRODUCIBILITY STATEMENT

To ensure the reproducibility of our work, we have provided comprehensive implementation details in both the main text and appendix of this paper. We study the effect of preference alignment on the widely-used LLaVA 1.6 Liu et al. (2024) model, and we exclusively examined publicly available preference datasets for alignment purposes. For the alignment methods we encourage the readers to follow the TRL library (https://github.com/huggingface/trl/), making modifications only when necessary to accommodate multi-modal inputs or online versions. All prompts used for different models are included in the appendix. By adhering to these practices, we aim to facilitate easy reproduction and validation of our results by the research community.

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

## A  IMPLEMENTATION DETAIL

For all offline experiments, as well as for Online-DPO and Mixed-DPO, we conducted a hyper-parameter search. The parameters included learning rates of $10^{-7}, 5 \cdot 10^{-7}, 10^{-6}, 5 \cdot 10^{-6}$; projection layer learning rates of $2 \cdot 10^{-5}, 2 \cdot 10^{-6}, 2 \cdot 10^{-7}$; epochs of 3, 5, and 7; and batch sizes of 16, 32, and 48. We reported the best results for each method. Additionally, we set the LoRA rank and scaling factor to 128 and 256, respectively. The $\beta$ values for DPO, IPO, and SLiC were explored at 0.05, 0.1, 0.2, 0.5 for DPO; 0.8, 0.9, 1.0 for IPO; and 0.02, 0.1, 0.2 for SLiC.

For RL methods (PPO and RLOO), we maintained constant base model parameters while training LoRA adapters for alignment. Specifically, for RLOO, we utilized $k = 4$, generating four distinct responses for each prompt at a temperature of 1.0. Training was conducted over two epochs with a batch size of 256 and a learning rate of $3 \cdot 10^{-4}$. Prior to RLOO training, we calculated the mean and standard deviation of rewards using the alignment dataset and normalized the rewards during training to achieve zero mean and unit variance. We determined that a $\beta$ value of $0.4$ provided the best balance between rewards and the KL penalty for RLOO. Gradient clipping was also implemented to cap the maximum gradient norm at 1.0.

For PPO specifically, we trained for 3 epochs with a learning rate of 1.41e-5 using a constant learning rate schedule. We used 1 GPU with a batch of 32. For the reward model, we used a a learning rate of 2e-5 and trained for 4000 steps. The learning rate schedule was also adjusted to be constant but with a warmup phase. The fraction value for the warmup phase is set at 0.03. Training was conducted on 8 GPUs with a batch size of 32.

For BDHS, we found that $\rho_{\text{th}}$ values close to 1 empirically gave the strongest results in our experiments with LLaVA 1.6. Therefore final results are reported at $\rho_{\text{th}} = 0.99$ if not otherwise stated. We argue that this is likely a result of the "AnyRes" technique used in LLaVA 1.6, which leads to significant redundancy across image tokens.

## B  EVALUATION

### B.1  EVALUATION BENCHMARKS

**Benchmarks**  We adopt multiple benchmarks to assess the capabilities of MLLMs, centered around both measuring the models visual faithfulness, i.e. its tendency to hallucinate, as well as overall helpfulness, i.e. the overall quality of its responses. Results have been obtained using an internal fork of lm-eval-harness (Gao et al., 2023b; McKinzie et al., 2024; Li et al., 2024).

`LLaVABench-in-the-Wild` (Liu et al., 2023b), `TextVQA` (Singh et al., 2019), and `GQA` (Hudson & Manning, 2019) help measure the model helpfulness, i.e. the effectiveness at following instructions and the completeness of the responses. `LLaVABench-in-the-Wild` expects free-form answers while both `TextVQA` and `GQA` require concise responses. We additionally report `MMVet` (Yu et al., 2023c), which evaluates the knowledge and visual reasoning capabilities of the MLLM. Such capabilities are not a direct target for most MLLM alignment strategies to improve. Nevertheless, `MMVet` offers a useful indicator for ensuring that such capabilities are not lost due to a possibly too simple or not sufficiently diverse alignment regiment.

`POPE` (Li et al., 2023b) and `MMHALBench` (Sun et al., 2023) evaluate the visual faithfulness of a model by identifying hallucinations in model responses. For `POPE`, we noticed that most of our experiments would reach a seeming plateau between 86% and 88% despite improvements in the other benchmarks. We conducted an initial manual review of 100 reported losses and observed incorrect or disputable ground truth on as many as 20 % of those samples (see Appendix B.2). While re-annotating those examples is beyond the scope of this work, we invite the community to consider it as many recent SOTA models exhibit such plateau[2].

Additionally, we noticed unexpected results on `MMHALBench`, and subsequent analysis showed limitations in its scoring. Specifically, `MMHALBench` uses text-only GPT-4 to detect hallucinations by comparing model responses to a reference response and a short list of objects known to be in the image. Sometimes this leads to entirely correct model responses to be marked as hallucinations

---

[2]See Table 4 in McKinzie et al. (2024) where all the models reported are demonstrating a plateau on `POPE`.

when they included more detail than the provided references. To mitigate this issue, we introduce a straightforward derivative we call `MMHALBench-V(ision)`, which relies on GPT-4o, i.e. provides the input image as additional context to the judge, to more reliably evaluate model capabilities. Data and evaluation prompts are unchanged. We empirically found this to be more reflective of true hallucinations in a human comparison. See Appendix B.4 for our review. Throughout experiments, we mainly focus on `MMHALBench-V` numbers and report `MMHALBench` primarily for reference.

While responses that have fewer hallucinations are often also inherently more helpful, we observe that these dimensions are nevertheless distinct and optimizing for reduction in hallucination crucially does not necessarily imply a more helpful model. In fact, in some instances, we even observed an inverse relationship. For example, as discussed in (Zhu et al., 2023), a given model would be more likely to hallucinate when asked to produce longer captions than shorter ones. This implies that models could learn to hallucinate less simply by providing more concise, arguably less useful, responses, and that models that aim to provide more detailed responses may find it more difficult to remain faithful to visual context in all respects[3]. For this reason, we report the recall metric from `Object HalBench` Yu et al. (2023b), styled Recall$^{coco}$ in our tables. This measures how many objects known to be in an image based on CoCo annotations are mentioned in a comprehensive caption given by the model. We considered as well reporting the `CHAIR` (Rohrbach et al., 2018) metrics from `Object HalBench` (Yu et al., 2023b). However, during our experiments, we found that those measurements were not always correlated with the quality of the models evaluated (see Appendix B.3).

## B.2 POPE

We noticed the existence of an upper bound on the `POPE` benchmark, as most of our experiments would reach a plateau between 86% and 88% despite improvements on other benchmarks. We manually looked at the losses among 100 responses and present the results in this section.

In 20% cases, we observed that the ground truth was either incorrect or disputable. In some of those cases, it appeared that the ontology used to build `POPE` could potentially result in differing interpretations. For example, in certain countries, a clear distinction exists between a car and a truck, although this distinction is not as pronounced in other regions of the world[4]. We provided an example along the response of our aligned model[5] in Figure 2.

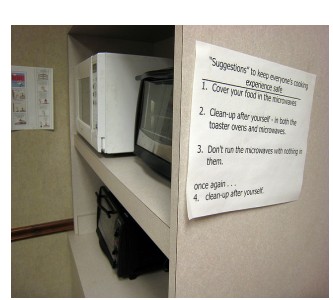

**Prompt:** Is there a tv in the image?
**POPE Ground Truth:** yes
**Aligned LLaVA 1.6-7B response:** no

Figure 2: Upon analysis of the losses on `POPE`, we noticed close to 20% of cases where the ground truth was either incorrect or disputable. This example is from `POPE`, which sources images from COCO (Lin et al., 2014).

Provided these examples are eliminated, we think it is plausible that performant models could potentially exceed a 90% accuracy rate on the `POPE` benchmark. Re-annotating those examples is beyond the scope of this work, however we would like to invite the community to consider it as

---

[3]To some extent, one could argue this mirrors the tension between helpfulness and safety as reported in Touvron et al. (2023), where a highly safe model may be less helpful.

[4]An example of such distinction between car/truck can be seen on COCO_val2014_000000210789.jpg where the `POPE` ground truth expects "no" to the prompt "Is there a car in the image?".

[5]We used a LLaVA 1.6-7B DPO-aligned on LLaVA prompts and responses sampled from VLFeedback. See Section 4.3.

many recent SOTA models exhibit such plateau. See Table 4 in (McKinzie et al., 2024) where all the models reported are demonstrating such plateau on `POPE`.

### B.3 CHAIR AND OBJECT HALBENCH

We evaluated two widely used benchmarks in the community for measuring hallucination, focusing on the computation of `CHAIR` metrics. We investigated approaches described by Rohrbach et al. (2018), which uses COCO annotations to compute `CHAIR` scores, and the more recent method by Yu et al. (2023b), named `Object HalBench`, which combines COCO annotations with a GPT model to enhance the detection of hallucinated objects.

Our analysis reveals that both benchmarks are significantly noisy (Figure 3). We also found that any improvements in `CHAIR` scores strongly depend on the ability of these benchmarks to detect specific types of hallucinations and cannot be attributed solely to the improvement of the model.

Furthermore, it is common to report `CHAIR` metrics without including recall metrics. Considering the trade-off between `CHAIR` and recall, omitting recall does not provide a full picture of how much a model has improved in reducing hallucinations. For instance, a model that generates short and conscise responses might not produce many hallucinations, but this may be at the cost of potentially providing an unhelpful answer.

Hence, the recall metric from Rohrbach et al. (2018) proves particularly informative for comparing different models and helping with our understanding of other benchmarks. We report this metric in our evaluations, styled $Recall^{coco}$ in our tables.

### B.4 MMHALBENCH-VISION

The original `MMHALBench` benchmark (Sun et al., 2023) uses GPT-4 to judge whether model responses introduce hallucinations. In that text-only regime, `MMHALBench` relies on ground truth information about the pictures, such as the categories of the objects present or a human reference response to the prompt.

We evaluated manually the common wins and losses obtained on `MMHALBench` during our experiments and noticed that in ∼20% cases we disagree with the resulting `MMHALBench` score[6]. We found cases where responses with hallucinations were considered as correct. Oppositely, we found cases where valid answers were wrongly tagged as containing hallucinations. In many cases, we saw the helpfulness to be under-estimated. See Figure 4.

This can be explained due the ground truth information being only expressed through text causing the judge model, GPT-4, to wrongly tag or miss hallucinations. To mitigate such cases, we introduced `MMHALBench-Vision`: we rely on the recently introduced GPT4-o to consume the image along the text ground truth information. We kept the evaluation prompt and scoring identical.

## C RL-BASED ALIGNMENT

To evaluate RL-based alignment methods we followed the established recipe of training a reward model on a preference dataset and then using an RL algorithm to optimize the MLLM to maximize the reward of responses sampled from the policy. We chose PPO and RLOO due to their popularity in the LLM literature.

**Reward Model Training and Evaluation**  We analyze the utility of datasets available in the community for reward model training by training on POVID, RLHF-V and VLFeedback preference datasets. To evaluate such created reward models in isolation, we hold out a small validation set split from the original dataset and report classification accuracy of the trained reward model, i.e. its ability to differentiate the chosen from the rejected response in POVID, RLHF-V, and VLFeedback sets. These held out validation sets are not used for reward model training.

Table 7 shows the performance of the reward models trained on different datasets across all validation sets. The model trained on VLFeedback shows the best generalization across the different datasets,

---

[6]21 cases out of 96 while comparing wins and losses of two models.

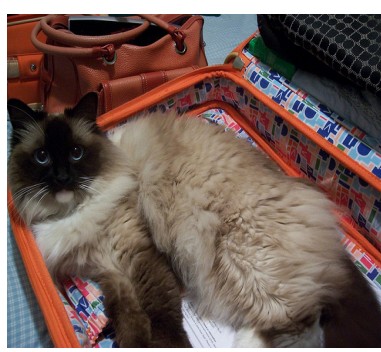

**Model Response:** The image features a brown and white cat lying on its side in an orange suitcase. The cat appears to be relaxed and comfortable in the suitcase. The suitcase is placed on a bed. The suitcase is open, revealing its contents, which include a book and a handbag. The cat is the main focus of the image, occupying a significant portion of the frame. The orange suitcase and the handbag are the other main objects in the scene, with the handbag being located on the left side of the image and the suitcase occupying the right side. The overall mood of the image is calm and cozy.

**Model Response:** The image features a large, colorful truck parked on the side of a street. The truck is parked next to a brick building, and there is a car parked behind it.
There are several people in the scene, with one person standing near the truck and two others further away. One of the individuals is carrying a handbag, which is visible in the scene. The overall atmosphere of the image is lively and engaging, with the eye-catching truck.

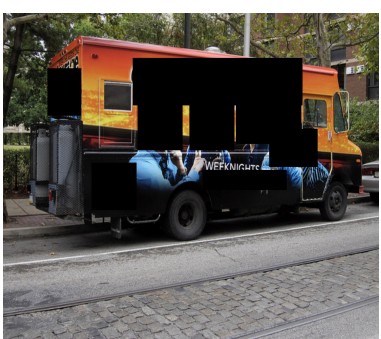

Figure 3: Examples illustrating instances where the `CHAIR` and `Objet HalBench` benchmarks fail to detect hallucinations. Text highlighted in green identifies hallucinations successfully detected by the benchmarks. In contrast, text highlighted in red indicates examples where the benchmark failed to identify hallucinations. Orange indicates hallucinations that, though not targeted by these benchmarks, degrade response quality. The top example shows the benchmark proposed by Yu et al. (2023b) while the bottom example follows from (Rohrbach et al., 2018). Images are from COCO (Lin et al., 2014).

| Base Model | Train Dataset | Held-Out Eval Dataset | | |
|---|---|---|---|---|
| | | POVID | RLHF-V | VLFeedback |
| LLaVA 1.5-7B | POVID | 0.99 | 0.24 | 0.56 |
| LLaVA 1.5-7B | RLHF-V | 0.12 | 0.86 | 0.52 |
| LLaVA 1.5-7B | POVID + RLHF-V | 0.98 | 0.76 | 0.53 |
| LLaVA 1.5-7B | VLFeedback | 0.61 | 0.54 | 0.81 |
| LLaVA 1.6-7B | POVID | 0.99 | 0.34 | 0.59 |
| LLaVA 1.6-7B | POVID + RLHF-V | 0.97 | 0.68 | 0.63 |
| LLaVA 1.6-7B | VLFeedback | 0.76 | 0.53 | 0.82 |

Table 7: Reward model accuracy on the held-out validation set.

as may be expected given its significantly larger size and higher diversity. In contrast, reward models trained on POVID and RLHF-V show notably poor generalization to their respective counterpart, while achieving high scores on their own held out portions. We hypothesize that the reward model may learn to recognize and prefer the respective (original) policy response before corruption (POVID) or enhancement (RLHF-V), which could explain the performance being significantly below a random choice baseline. To strengthen our hypothesis, we also combine the POVID and RLHF-V sets for reward model training and observe that both LLaVA 1.5-7B and LLaVA 1.6-7B are able to learn a more balanced objective, although even for such a combined training set we still observe limited generalization to VLFeedback.

| image ID: 16189396430_4dce91a9d7_o | **Prompt:** How many people are there in the image?
**MMHAL reference response:** There are four people in the image.
**MMHAL categories**: Girl, Human head, Human hair, Mammal, Dog, Person, Human face
**LLaVA 1.6-7B aligned:** There are four people in the image: a man, a woman, and two children.

**GPT-4 response:** total score=1, hallucination=1, helpfulness=1 |
|---|---|

| image ID: 12392564684_ffd19b5b55_o | **Prompt :** Compare the two different views of the backpack.
**MMHAL reference response:** The image shows two different views of the same black backpack. On the left side, it shows the inside of the bag, containing a camera, two lenses an other accessories. On the right side, it shows the outside of the bag, and there is a "Lowepro" logo on it.
**MMHAL categories**: Hiking equipment, Clothing, Backpack
**Aligned LLaVA 1.6-7B response:** The image shows a backpack with two different views: the top view and the side view. In the top view, we can see (...)

**GPT-4 response:** total score=4, hallucination=0, helpfulness=1 |
|---|---|

Figure 4: Example of incorrect assessments with `MMHALBench`. The top example shows a response incorrectly assessed as having a hallucination. The second presents a response where the model hallucinates a side view of the backpack, but that is missed by `MMHALBench`. In both cases, the text ground truth does not give enough information to the text-only evaluator to evaluate the response. Reproduction of `MMHALBench` images cannot take place here due to usage restrictions. Please refer to `https://huggingface.co/datasets/Shengcao1006/MMHal-Bench`.

**RL Training and Evaluation** We used the POVID and VLFeedback based reward models for PPO and RLOO training. Table 8 shows the scores of the best models trained via PPO and RLOO.

| Alignment | Dataset$_{RM}$ | Dataset$_P$ | POPE↑ | MMHAL↑ | MMHAL$^V$ ↑ | LLaVA$^W$↑ | VQA$^T$ ↑ | GQA↑ | MMVet↑ | Recall$^{coco}$ ↑ |
|---|---|---|---|---|---|---|---|---|---|---|
| Baseline | – | – | 86.40 | 2.95 | 2.75 | 80.85 | 64.85 | 64.23 | 43.94 | 68.13 |
| DPO | – | POVID | 88.09 | 3.16 | 3.07 | 78.05 | 64.56 | 64.12 | 40.60 | 73.48 |
| PPO | POVID | POVID | | | | | | | | |
| RLOO | POVID | POVID | | | Policy training not stable | | | | | |
| PPO | VLFeedback | POVID | 87.54 | 3.02 | 3.09 | 80.17 | 63.90 | 64.04 | 40.51 | 67.19 |
| RLOO | VLFeedback | POVID | 87.17 | 2.94 | 2.72 | 78.72 | 63.59 | 63.72 | 42.25 | 64.57 |

Table 8: RL-based alignment of LLaVA 1.6-7B, DPO baseline included for reference. RL-based alignment methods use a reward model based on LLaVA 1.6-7B, Dataset$_{RM}$ refers to the dataset used to train the reward model, Dataset$_P$ to the set of images and prompts used for RL alignment.

Mirroring the observed lack in generalization in our reward model experiments, we found that using POVID-based reward model resulted in collapse of responses during the RL training. Only the use of the reward model trained on the much larger VLFeedback dataset allowed for stable RL training without model collapse. We hypothesize that besides the larger size, VLFeedback may be more aligned with the downstream objective of the reward model due to its construction by ranking sampled model responses, compared to POVID, which aims to produce minimally different preference pairs. Nevertheless, even the stronger VLFeedback-based reward model did not allow us to reliably outperform a much simpler DPO baseline[7].

These observations indicate that reward model training with subsequent RL alignment could perhaps require more carefully curated data, e.g., with more focus on diversity, than direct alignment methods where both POVID and VLFeedback individually achieve strong improvements. In addition to inherently stronger reward models, perhaps basing them on more powerful base models, it also suggests that the approach introduced in the concurrent work of Yu et al. (2024), which introduces a

---

[7]We also found that models achieving higher reward during RL training, did not perform better than models with lower reward and less KL divergence, i.e., models with higher $\beta$ parameter performed better on the benchmarks. None of the RL algorithms clearly outperformed the others.

symbolic reward formulation based on scores from a VQA model verifying statements made by the policy may be a promising avenue for future research.

Another interesting observation is that the RL aligned models show similar evaluation trends as the DAP aligned models, where both use POVID prompts and images for the training of the policy. For example, compared to the base model they show some improvement in `POPE`, and `MMHalBench` (both variants), with some regressions in `LLaVABench-in-the-Wild`, `TextVQA`, `GQA`, and `MMVet`. These trends are distinct to what is seen when using direct preference alignment on VLFeedback data as shown in Table 3. This is remarkable as the RL aligned models do of course not use the chosen and rejected responses present in the POVID dataset, instead getting their feedback signal entirely from the reward model which is trained on VLFeedback data. We observe a similar trend in Section D.3, where in a purely online setting, the choice of input prompts and images significantly impacts alignment results.

# D    ONLINE DPO

## D.1    ANNOTATOR IN ONLINE-DPO

Table 9 shows the prompt we used to obtain online feedback from the annotator. We conducted multiple experiments with different prompts. In one setup, similar to the approach taken by Guo et al. (2024) with the rewards model, we included the ground truth response as an additional signal for the annotator to evaluate both responses. We did not observe any significant change in either the evaluation metrics or the final performance of the aligned model. This may be due to the fact that most of the open-source MLLMs we used in this study still lack the ability to follow instructions effectively, especially when the instructions contain multiple components or detailed steps.

We also examined the potential bias of the annotator model in choosing "Response 1" or "Response 2" and found no noticeable bias. Figure 5 shows an example of an annotation made by LLaVA 1.6-34B model.

> The most important part of this task is to choose a response that contains less hallucination. Everything in the answer should be based on the contents of the image. You are given an image, a question, and two responses. If the context is about something practical, a helpful response is a concise response, and not one with irrelevant questions and comments. You are an expert annotator, and you rate the answer with less hallucination and more helpful information about the image as the better answer. Less hallucination means every object or attribute of the object, like color and relationship, is described accurately and as it appears in the image. If something is ambiguous in the image, the answer should avoid including any details that are not clearly visible in the image. In your response, you should generate an answer where you indicate whether Response 1 or Response 2 is better and explain the reason.

Table 9: The Prompt Used for the Annotator.

## D.2    HOW DOES A STRONGER ANNOTATOR AFFECT THE PERFORMANCE OF ALIGNED MODEL?

**Annotator Evaluation** We used LLaVA 1.6-34B as the annotator. To verify its capability to accurately judge different responses, we evaluated it on the sample held-out part of three datasets we used for evaluating the reward model in Section C. Results are summarized in Table 10. For further details on the prompts used and qualitative examples of the annotator's outputs, please refer to Appendix D.1.

| Dataset | LLaVA 1.6-7B | LLaVA 1.6-34B |
|---|---|---|
| VLFeedback (eval) | 79.10 | 90.91 |
| RLHF-V (eval) | 81.88 | 93.90 |
| POVID (eval) | 92.96 | 98.55 |

Table 10: Performance of the annotators on different preference datasets.

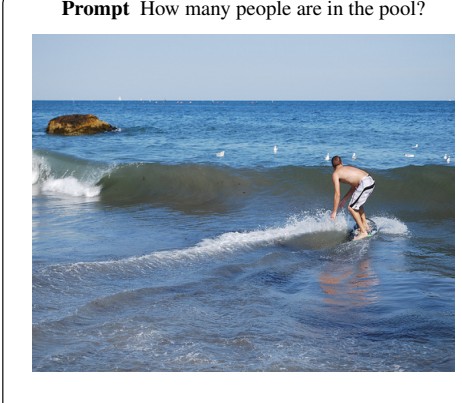

**Prompt** How many people are in the pool?

**Response 1:** There is a person in the pool, a young man enjoying his time surfing the small waves.

**Response 2:** This is not a swimming pool but the ocean, and on the ocean, there is a person. A young man is enjoying his time surfing.

**Annotator:** Based on the image content, Response 2 is better and more helpful. The image shows a person surfing in the ocean, not a swimming pool. The surfer is riding a wave, and there are no other people visible in the image. The description accurately reflects the content of the image and does not include any hallucinations or irrelevant details.

Figure 5: Example of Annotation. Image is from COCO (Lin et al., 2014).

Table 11 presents a comparison of the effects of Online-DPO with two different annotators.

While using Online-DPO with LLaVA 1.6-7B as the judge can enhance the overall performance of the model, the stronger annotator seems to provide more consistent improvements across various benchmarks.

Concurrent to us, Yu et al. (2024) proposed an annotation approach that segments the annotation process into easier sub-tasks, with each task being individually scored. These scores are then aggregated to form an overall score that rates the responses. This method can potentially enable weaker models to still provide strong supervision signals during the alignment process. Moreover, exploring the use of stronger base models and diverse datasets, both in terms of size and variety, could further enhance the effectiveness of the online approach. We leave the detailed investigation of these aspects for future work.

| Model | Dataset | Annotator | POPE ↑ | MMHAL↑ | MMHAL$^V$ ↑ | LLaVA$^W$ ↑ | VQA$^T$ ↑ | GQA ↑ | MMVet ↑ | Recall$^{coco}$ ↑ |
|-------|---------|-----------|--------|--------|-------------|-------------|-----------|-------|---------|-------------------|
| LLaVA 1.6-7B | – | – | 86.40 | 2.95 | 2.75 | 80.85 | 64.85 | 64.23 | 43.94 | 68.13 |
| LLaVA 1.6-7B | POVID | LLaVA 1.6-7B | 86.54 | 2.52 | 2.72 | 81.55 | 64.93 | 64.18 | 40.73 | 67.40 |
| LLaVA 1.6-7B | POVID | LLaVA 1.6-34B | 86.49 | 2.88 | 2.94 | 82.61 | 64.88 | 64.31 | 43.26 | 68.45 |

Table 11: Comparison of Online-DPO with a strong annotator (i.e., LLaVA 1.6-34B) and a weak annotator (i.e., LLaVA 1.6-7B).

### D.3 ONLINE-DPO & MIXED-DPO

We apply Online-DPO and Mixed-DPO to both the POVID and the RLHF-V dataset. The results are summarized in Table 12. Consistent with our observations on the POVID dataset, applying Mixed-DPO – which combines elements of DPO and Online-DPO – typically results in a moderating effect on performance outcomes. The results often span a range slightly broader than the highest and lowest performances achieved by DPO and Online-DPO. This variability is attributed to the probabilistic nature of the online sampling in Online-DPO.

On the RLHF-V dataset, where Online-DPO consistently outperforms DPO across all benchmarks, the moderating effect of Mixed-DPO proves not beneficial, as the offline DPO component contributes minimally to the overall model performance. Nevertheless, Mixed-DPO remains a valuable strategy in scenarios where, as observed in the experiments on the POVID dataset, offline and Online-DPO show complementary improvements, leveraging the strengths of both to optimize overall performance.

| Alignment | Dataset | POPE ↑ | MMHAL↑ | MMHAL$^V$ ↑ | LLaVA$^W$ ↑ | VQA$^T$ ↑ | GQA ↑ | MMVet ↑ | Recall$^{coco}$ ↑ |
|---|---|---|---|---|---|---|---|---|---|
| – | – | 86.41 | 3.06 | 2.71 | 78.96 | 64.22 | 64.22 | 43.94 | 68.13 |
| DPO | POVID | 88.09 | 3.16 | 3.07 | 78.63 | 64.56 | 64.12 | 40.60 | 73.48 |
| Online-DPO | POVID | 86.49 | 2.88 | 2.94 | 82.61 | 64.88 | 64.31 | 43.26 | 68.45 |
| Mixed-DPO | POVID | 88.03 | 2.83 | 3.10 | 82.75 | 64.93 | 64.47 | 42.80 | 74.53 |
| DPO | RLHF-V | 83.86 | 3.15 | 3.26 | 70.58 | 64.75 | 62.89 | 37.16 | 64.26 |
| Online-DPO | RLHF-V | 85.40 | 3.10 | 3.27 | 79.66 | 64.94 | 64.05 | 41.01 | 68.13 |
| Mixed-DPO | RLHF-V | 85.57 | 2.94 | 3.16 | 78.46 | 65.06 | 64.10 | 41.10 | 67.82 |

Table 12: The effect of Mixed-DPO, using LLaVA 1.6-7B as the base model.

# E    COMPARISON OF DIFFERENT OFFLINE ALIGNMENT METHODS

While we conducted most of our experiments using DPO for comparability with other works in the community, we also ran a few experiments to investigate whether other popular offline methods could improve the results. Results are summarized in Table 13.

Our results indicate that both IPO and SLiC, similar to DPO, boost the model's performance across most hallucination benchmarks. Additionally, these methods demonstrate improvements in more open question-answering benchmarks. We anticipate that Online-IPO and Online-SLiC will yield enhancements over their offline counterparts — similar to the improvements observed with Online-DPO over DPO — as examined in Guo et al. (2024). However, this study is beyond the scope of this paper and is left for future work. Primarily, we aim to highlight the importance of considering different alignment objectives, emphasizing that the choice between offline objectives in different setups can impact the effect of the alignment pipeline.

| Alignment | Dataset | POPE ↑ | MMHAL↑ | MMHAL$^V$ ↑ | LLaVA$^W$↑ | VQA$^T$ ↑ | GQA ↑ | MMVet ↑ | Recall$^{coco}$ ↑ |
|---|---|---|---|---|---|---|---|---|---|
| – | – | 86.40 | 2.95 | 2.75 | 80.85 | 64.85 | 64.23 | 43.94 | 68.13 |
| DPO | POVID | 88.09 | 3.16 | 3.07 | 78.63 | 64.56 | 64.12 | 40.60 | 73.48 |
| IPO | POVID | 87.62 | 3.11 | 3.11 | 82.34 | 65.09 | 64.47 | 43.99 | 69.81 |
| SliC | POVID | 88.28 | 3.17 | 3.15 | 81.99 | 64.59 | 64.11 | 41.51 | 74.32 |

Table 13: Comparison of different offline alignment methods based on LLaVA 1.6-7B.

# F    PREFERENCE DATA

## F.1    STATE OF THE ART PREFERENCE DATASETS

This section presents recently published multimodal preference datasets. We categorize those contributions into two categories according to their annotation source:

**Human annotations**  In LLaVA-RLHF (Sun et al., 2023), authors collect human preferences by asking crowdworkers to prioritize responses that minimize hallucinations. Using this process, the authors built a 10k preferences dataset. The prompts are from LLaVA-Instruct-150k (Liu et al., 2023c), while responses are sampled from LLaVA base model. As is common for many other preferences datasets, the source of the images is COCO (Lin et al., 2014).

In RLHF-V, Yu et al. (2023b) collect human preferences at the segment level by asking annotators to correct mistakes in model responses. Used in conjunction with a token-weighted DPO training, authors reported a reduced hallucinations level. Prompts and images are originally from the UniMM-Chat SFT dataset introduced by Yu et al. (2023a), and responses sent for annotation are sampled from Muffin (Yu et al., 2023a).

**Synthetic annotations**  In DRESS (Chen et al., 2023), authors introduce NLF, a 63k pairwise preference dataset built from LLaVA-Instruct-150k images and prompts. Authors leverage GPT-4 to provide critique and refinement on the responses of their in-house DRESS model. In VLFeedback (Li et al., 2023a), authors sample responses from a pool of 12 multimodal MLLMs — including GPT-4V, the LLaVA 1.5 series models and Qwen-VL-Chat— on a pool of prompts datasets (LLaVA-Instruct-150k, SVIT (Zhao et al., 2023a) (Visual Genome images), LLaVAR (Zhang et al., 2023) (LAION-5B

images)). This synthetic approach scales up both the number of examples generated and the diversity of the responses. GPT-4V is used to select the best responses.

In POVID, Zhou et al. (2024) generate dispreferences from a ground truth dataset directly, removing the need for ranking responses. Specifically, 17k examples are selected randomly from the LLaVA-Instruct-150k dataset, with the original answers assumed to be a preferred response, while the dispreferred response is derived by prompting GPT-4V to introduce mistakes in the preferred response.

**Preference signal** Transversally to the source of annotations, we consider how the preference signal is composed. LLaVA-RLHF (Sun et al., 2023) and VLFeedback Li et al. (2023a) use **ranking**: responses are sampled, then ranked by humans or GPT-4V. Other works use a **construction** approach where the signal is obtained by correcting responses, such as RLHF-V (Yu et al., 2023b) and DRESS (Chen et al., 2023), or by corrupting the responses such as POVID (Zhou et al., 2024).

### F.2 Dataset Prompt Corruption

We leverage GPT-4 to corrupt chosen responses with realistic and plausible hallucinations. We call *realistic hallucinations* those instances where a human, just by looking at the corrupted response, is unable to recognize it without having to refer back to the image. We have remarked this was an important distinction: the more obvious the corruptions are, the poorer the performance of the resulting policy is. We launched side experiments where we employed a less skilled LLM corrupter and incorporated obvious tags[8] into the responses. In both scenarios, we noticed a drop in performance as the corruption gets less realistic and readily 'hackable' by the policy under alignment. The prompt used to corrupt the chosen responses is reproduced in Table 14.

### F.3 Dataset Size ablation with the Corrupting Strategy

We conducted a dataset size ablation on the application of our corrupting strategy on VLFeedback (Figure 6) . We evaluated 7 checkpoints between 100 and 5,000 training samples, our maximum in this data regime (Section 4.3). We provide the baseline results with a dashed line. While POVID shows the best result on Recall$^{\text{coco}}$, our simple corruption strategy applied outperforms other datasets on both `LLavaBench-in-the-Wild` and `MMHALBench` hallucination rate, while being on par on the `MMHALBench` helpfulness rate with VLFeedback vanilla.

## G Bias-Driven Hallucination Sampling

This section provides additional details and supplemental results for the proposed bias-driven hallucination sampling (BDHS) approach.

### G.1 Related Work

As part of the POVID work, Zhou et al. (2024) suggests to trigger inherent hallucination patterns directly by presenting noisy images $\tilde{x}_{\text{img}}$ to the model when generating the non-preferred response $\tilde{y}^-$. Hereby, each generated token $t$ in $\tilde{y}^-$ is conditioned on the prior tokens from the preferred response $y^+_{<t}$, i.e., $\pi_\theta(\tilde{y}^-_t | \tilde{x}, y^+_{<t})$ with modified input $\tilde{x} = (x_{\text{text}}, \tilde{x}_{\text{img}})$ (teacher-forcing). The tilde notation emphasizes that the response is driven by the model with restricted access to the image. $\tilde{x}_{\text{img}}$ is created through a diffusion process that incrementally adds Gaussian noise to the image $x_{\text{img}}$ for a predefined number of steps $N$, which is set to 500 by default (see Section G.2 for implementation details).

The online response $\tilde{y}^-$ is combined with existing preference pairs $(y^+, y^-)$ by averaging pairwise losses[9], i.e.,

$$L_{\text{Avg-DPO}}(\pi_\theta) = \mathbb{E}_{\substack{(x,y^+)\sim D \\ \tilde{y}^- \sim \pi_\theta(\tilde{x}_{\text{img}}, y^+)}} \left[ \gamma L_{\text{DPO}}(y^+, y^-, x; \pi_\theta) + (1-\gamma) L_{\text{DPO}}(y^+, \tilde{y}^-, x; \pi_\theta) \right] , \quad (5)$$

---

[8]An example of obvious rejected response marked with a tag: *There are only two examples of plates that are visible in the image.* **This response is factually incorrect.**

[9]The loss presented here refers to the version in the public source code of POVID (Zhou et al., 2024) which averages the individual DPO losses, opposed to the paper (Zhou et al., 2024) that combines the non-preferred responses by a weighted sum of their log-probabilities.

You are provided with a prompt and a response to that prompt. You task is to corrupt the response with realistic hallucinations.

Hallucinations can be including objects/persons/animals not present in the original response, changing the positions of the objects/persons/animals, changing the count, changing the attributes (e.g. color, shapes, etc.) or their spatial relationships. Those hallucinations have to be realistic.

In other words, your task is to change the provided response in a way that could be correct given the prompt, but is not. A person only seeing the corrupted response shall not be able to identify it as being incorrect, so your change must be a plausible (although again incorrect). Output only the updated, corrupted response, nothing else.

Here are a few examples:
### Example 1
prompt: What is the title of the book mentioned in the image?
response: The title of the book in the image is "Picture This: A Gallery of Fusible Appliqué Projects for Quilting and Framing" by Jean Wells & Marina Anderson.
corrupted response: The title of the book in the image is "Sewing Masters: A Collection of Innovative Embroidery Designs for Quilting and Framing" by Marvin Wells & Vincent Anderson.

### Example 2
prompt: You are a driving assistant. Based on current image, what is the best action to take when you are driving on the road? A. Slow down B. Turn around C. Stop the car D. Turn Left E. Keep driving. Please give reasons and the answer
response: The best action to take when driving on the road, based on the current image, would be: E. Keep driving.
Reasons for this action:
1. The traffic light visible in the image is green, indicating that it is legal and safe to proceed.
2. The speed limit sign shows "60" which means the car, currently at 20 km/h, is well below the maximum speed limit and can safely continue and even increase speed if necessary.
3. The road ahead is clear of any immediate hazards or obstructions that would require stopping or turning.
corrupted response: The best action to take when driving on the road, based on the current image, would be: C. Stop the car.
Reasons for this action:
1. Although the traffic light on the left in the image is green, the one on your side is red, indicating it is necessary to stop
2. The speed limit sign shows "60" which means the car, currently at 20 km/h, is well below the maximum speed limit and can safely stop before the intersection.
3. The intersection up ahead indicates the presence of crossing cars, requiring a stop.

### Example 3
prompt: {original_prompt}
response: {original_response}
corrupted response:

Table 14: Prompt used to corrupt datasets with GPT-4.

with $\gamma = 0.5$.

Zhou et al. (2024) suggests that the proposed teacher-forcing strategy can help to yield samples that exhibit corruption only in few, key tokens most informed by the visual content, thus focusing the feedback signal for alignment. However, since the method operates token by token, we found that this can introduce non-sensical responses, e.g., only corrupting some parts of multi-token noun phrases. Such constructions would presumably already achieve a low generation probability, limiting the learning signal for the DPO-based alignment.

Concurrent to our BDHS work, (Yu et al., 2024) also emphasizes the significance of generating model samples with minimal differences. While their insights on annotation strategy are interesting, their proposed "Deconfounded Candidate Response Generation" approach appears similar to common

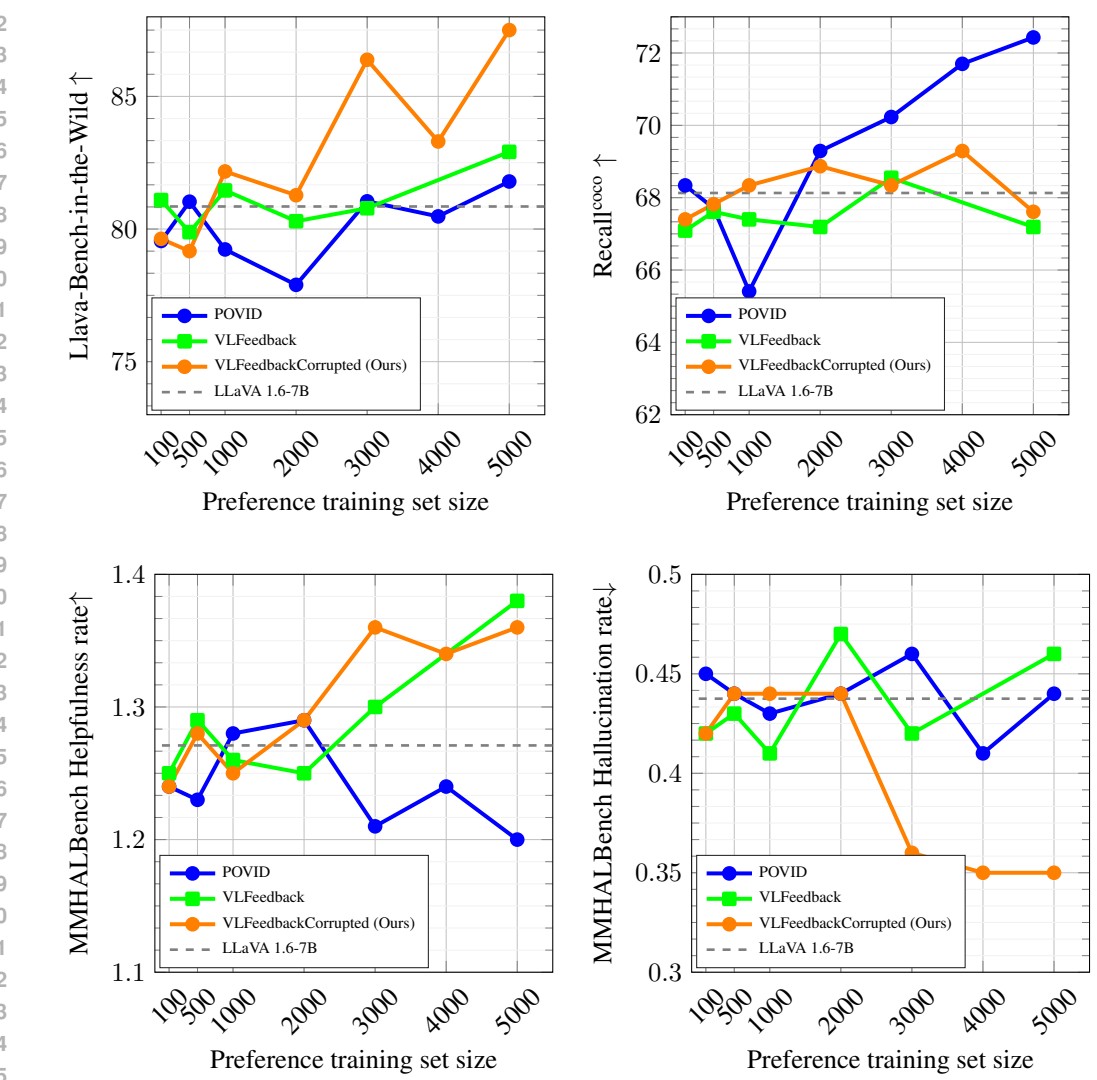

Figure 6: Impact of the preference dataset size. Our corrupting strategy outperforms other datasets on `LLaVA-in-the-Wild` and `MMHALBench` hallucination rate. It is on par on the `MMHALBench` helpfulness rate against vanilla VLFeedback. Finally, POVID reports the highest Recall$^{\text{coco}}$. The dashed lines are the scores for the LLaVA 1.6-7B baseline.

sampling techniques using higher temperatures in online pipelines, which do not necessarily create pairs of minimal differences. In another concurrent work, Deng et al. (2024) proposes generating "rejected responses" through image corruption. Despite the conceptual resemblance, we find that both using an attention mask and SFT-guided corruption are crucial in our final BDHS design (see Section 4.4).

## G.2 ADDING NOISE TO IMAGES

This section describes how to gradually add noise to images through a diffusion process. The derivation follows the public implementation of POVID-style image distortion (Zhou et al., 2024) to enable the proper reproduction of their results.

Let $x_{\text{img}}(k)$ denote the image after applying noise $k$-times with $x_{\text{img}}(0)$ referring to the original image and $\mathcal{N}(0, 1)$ represent the normal distribution. Then the forward noise process is defined as:

$$x_{\text{img}}(k) = \sqrt{1 - \beta_k}\, x_{\text{img}}(k - 1) + \sqrt{\beta_k}\epsilon \quad \text{with } \epsilon \sim \mathcal{N}(0, 1). \tag{6}$$

Hereby, $\beta_k$ denotes a time-variant parameter which is set to $\beta_k = \sigma(-6 + \frac{12k}{1000}) \cdot (0.5 \cdot 10^{-2} - 10^{-5}) + 10^{-5}$ to gradually increase noise between $k = 0$ and $k = 1000$ (refer to Figure 7).

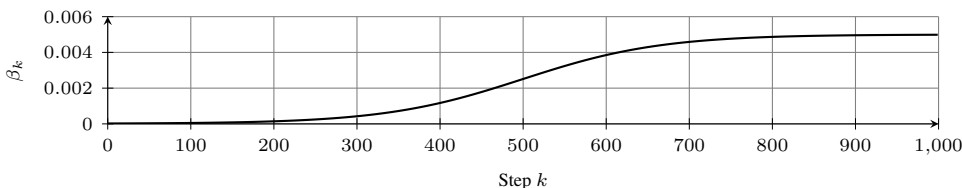

Figure 7: Schedule of diffusion parameter $\beta_k$.

The recursive equation equation 6 can be reformulated to apply $k$ steps of noise at once. Setting $\alpha_k = 1 - \beta_k$ and $\bar{\alpha}_N = \prod_{k=1}^{N} \alpha_k$, the following equation applies $N$ steps of noise to image $x_{\text{img}}(0)$:

$$\tilde{x}_{\text{img}}(N) = \sqrt{\bar{\alpha}_N}\, x_{\text{img}}(0) + \sqrt{1 - \bar{\alpha}_N}\epsilon \quad \text{with } \epsilon \sim \mathcal{N}(0, 1). \tag{7}$$

The default for $N$ in Zhou et al. (2024) is $N = 500$.

## G.3 BDHS: Reference-Guided Generation

This section provides more details about the reference-guided generation as summarized in Section 3.1. We assume that the preferred response $y^+$ can be split into $k = 1, 2, \ldots, S$ sentences with $y_k^+$ denoting the $k$-th sentence of $y^+$. Each sentence is decomposed into two parts $y_{k,1}^+$ and $y_{k,2}^+$, respectively, at a randomly sampled position, i.e., $y_k^+ = (y_{k,1}^+, y_{k,2}^+)$. The model $\pi_\theta$ is then invoked to generate a corresponding corrupted sentence $\tilde{y}_k^- = (y_{k,1}^+, \tilde{y}_{k,2}^-)$ whereas

$$\tilde{y}_{k,2}^- \sim \pi_\theta(\cdot | \tilde{x}, y_{<k}^+, y_{k,1}^+). \tag{8}$$

Note that this is an abuse of notation for better readability, as $\tilde{y}_{k,2}^-$ denotes the full response sampled from multiple model invocations until the first full stop or end of sequence token. Every sentence is based on the full ground truth from the previous sentence $y_{<k}^+$ and not the previously generated output $\tilde{y}_{<k}^-$ to improve consistency. Finally, the full BDHS response is given by concatenation of the individual sentences, i.e. $\tilde{y}^- = (\tilde{y}_1^-, \tilde{y}_2^-, \ldots, \tilde{y}_S^-)$. Note, the partitioning into sentences is a design decision to keep the non-overlapping portion between $y^+$ and $\tilde{y}^-$ reasonably small and to improve consistency when switching forth and back between responses. In the implementation, the generation of responses for several sentences and preference pairs can be highly parallelized as equation 8 does not depend on any previously generated output for all $k$.

For question answering tasks, several ground truth responses $y^+$ consist of only one or few words and often start with *yes* or *no*. In these cases $\tilde{y}_{k,2}^-$ can often easily inferred from $y_{k,1}^+$ even without image access at all and therefore we extend the previous strategy by a simple heuristic: whenever $y_{k,1}^+$ starts with a *yes* or *no* it is substituted by its counterpart with a probability of 50%.

## G.4 BDHS: Ensuring Semantically Meaningful Differences

Section 3.1 describes an iterative technique for BDHS that evaluates similarity scores between the generated response $\tilde{y}^-$ and the ground truth $y^+$. If both responses are identified as similar according to the sentence embeddings model, a new BDHS response is sampled until a maximum number of iterations $N_{\text{BDHS}}$ is reached. The last iteration waives the ground truth reference and generates a full response which is then taken as $\tilde{y}^-$ regardless of the similarity score. Figure 8 shows the number of non-similar responses, i.e. ensuring $\epsilon_s < 0.97$, over the number of BDHS iterations for the full POVID (5k) dataset. As expected all BDHS variants result in a larger number of non-similar responses

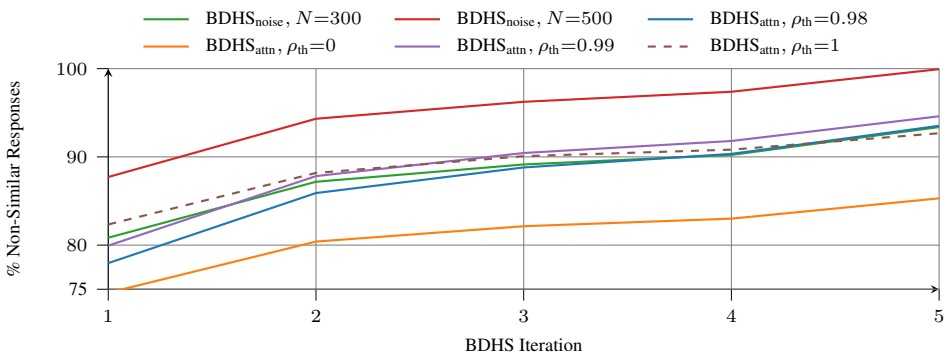

Figure 8: Number of resolved similar responses for BDHS generation based on POVID (5k). Parameters are $\epsilon_s = 0.97$ and $N_{\text{BDHS}} = 5$.

compared to the model response without image attention blocking or noisy images. Running BDHS with a single iteration already results in more than $74\%$ semantically different responses. After four iterations, BDHS variants with restricted image access differ in over $90\%$ while the last iteration is guidance free and only depends on the sampled attention mask resp. noise. Interestingly, in iteration 5, $\text{BDHS}_{\text{attn}}, \rho_{\text{th}}=1$ corresponds to guidance-free response generation with fully blocked image tokens which still results in $7\%$ similar responses w.r.t. the SFT ground truth. Probable reasons for this saturation are either that the correct answer is easy to guess even without access to the image, or that the answer is memorized from the training data. Note that prompts and images in POVID (5k) are extracted from LLaVA Instruct which served as training data for fine-tuning LLaVA-1.6.

BDHS with noisy images in the input and $N = 500$ diffusion steps results in more than $99\%$ semantically different responses after five iterations, surpassing the score for the fully blocked response. This is misleading, as although the responses are indeed semantically different, they mostly mention that the prompt cannot be evaluated due to blurry and noisy images. Essentially, the noise adds an additional bias towards noise/pixel-referring responses instead of inducing only the desired inherent bias which would saturate at approx. $93\%$ (response with fully masked image tokens).

After three iterations, the score of $\text{BDHS}_{\text{attn}}, \rho_{\text{th}}=0.99$ reaches the one from the fully blocked response which is hypothetically implied due to increased diversity by subsampling a distinct attention mask.

Section G.7 presents several examples with actual responses.

### G.5 BDHS ALGORITHM

The general overview of BDHS is provided in Figure 1. This section introduces the corresponding algorithm which is listed in Algorithm 1. This version includes both, noisy images for $\text{BDHS}_{\text{noise}}$ and attention masking for $\text{BDHS}_{\text{attn}}$ (refer to the comments in Algorithm 1). We add a straightforward heuristic to swap *yes* and *no* words whenever they occur in the beginning of a sentence. For this purpose line 12 introduce a regular expression which matches any *yes* or *no* at the beginning of each sentence and optionally skips any preceding newline or whitespace characters. This expression can be extended to further use-cases if desired. We choose to generate the full response without any SFT ground truth guidance in the very last iteration whenever $N_{\text{BDHS}} > 1$ to minimize similarity (refer to line 5).

### G.6 ADDITIONAL ABLATIONS

Additional BDHS ablations, especially regarding different hyperparameter choices are shown in Table 15. We also evaluate SFT guidance-free generation only with attention masking active. The corresponding benchmark results are listed in the first two rows. All subsequent rows evaluate the full BDHS approach including SFT guidance. We include ablations that rely on noisy images rather than attention masking, following the diffusion process described in G.2. Table 16 presents results for larger BDHS datasets, in particular for full POVID ($\approx$17k samples) and VLFeedback with GPT-4V responses ($\approx$34k samples, refer to Section 4.3 for details about the dataset modification).

---

**Algorithm 1** BDHS

---

**Require:** Prompt $x_{\text{text}}$, image $x_{\text{img}}$, SFT ground truth $y^+$, attention masking parameter $\rho_{\text{th}}$, image noise level $N$, BDHS iterations $N_{\text{BDHS}}$, similarity threshold $\epsilon_{\text{s}}$

1: **for** $i = 1, 2, \ldots, N_{\text{BDHS}}$ **do**
2:     $m \leftarrow$ Sample image attention mask with $\rho_{\text{th}}$ according to equation **??**     $\triangleright \rho_{\text{th}} > 0$ only for BDHS$_{\text{attn}}$
3:     $\tilde{x}_{\text{img}} \leftarrow \text{AddNoise}(N, x_{\text{img}})$ via equation 7     $\triangleright N > 0$ only for BDHS$_{\text{noise}}$
4:     $\tilde{x} \leftarrow (x_{\text{text}}, \tilde{x}_{\text{img}}, m)$
5:     **if** $N_{\text{BDHS}} > 1$ and $i = N_{\text{BDHS}}$ **then**
6:         $y^- \leftarrow$ Generate full model response via $\pi_\theta(\cdot|\tilde{x})$
7:         **return** $y^-$
8:     $\mathcal{S} \leftarrow$ Split $y^+$ into S sentences
9:     $y_k^- \leftarrow \emptyset$     $\triangleright$ Initialize empty string
10:     **for each** $y_k^+$ in $\mathcal{S}$ **do**     $\triangleright$ Parallelizable
11:         $\xi \leftarrow \xi \sim \mathcal{U}(0,1)$     $\triangleright \mathcal{U}(0,1)$ denotes the uniform distribution in $[0,1]$
12:         **if** $y_k^+$ matches $r''^\wedge[\backslash s]*(Yes|yes|No|no)''$ and $\xi \geq 0.5$ **then**   $\triangleright r''\cdot''$ denotes a regular expression
13:             $y_k^+ \leftarrow$ Swap corresponding *Yes/yes* by *No/no* and vice versa
14:         $y_{k,1}^+ \leftarrow$ Sample random position in $y_k^+$ and return first substring
15:         $y_{k,2}^- \leftarrow$ Complete sentence via equation 8 until full stop or *<eos>*
16:         $y_k^- \leftarrow (y_{k,1}^+, y_{k,2}^-)$     $\triangleright$ Concatenate strings to full sentence
17:         $y^- \leftarrow (y^-, y_k^-)$     $\triangleright$ Append to overall response
18:     $\phi \leftarrow$ Compute similarity score between $y^-$ and $y^+$ in $[0,1]$     $\triangleright$ Use sentence embeddings
19:     **if** $\phi < \epsilon_{\text{s}}$ **then**
20:         **break**     $\triangleright$ Semantically different according to threshold
21: **return** $y^-$

---

| $\tilde{y}^-$ derived from policy | POPE↑ | MMHAL↑ | MMHAL$^V$ ↑ | LLaVA$^W$↑ | VQA$^T$ ↑ | GQA↑ | MMVet↑ | Recall$^{\text{coco}}$ ↑ |
|---|---|---|---|---|---|---|---|---|
| – (Baseline) | 86.40 | **2.95** | 2.75 | 80.85 | 64.85 | 64.23 | **43.94** | 68.13 |
| – (Plain DPO) | 88.18 | 2.93 | **2.93** | 81.89 | 64.90 | 64.34 | 43.39 | 71.80 |
| Attention Masking, $\rho_{\text{th}} = 0.98$ | 88.61 | 2.25 | 2.25 | 82.25 | 64.92 | 64.04 | 42.75 | **77.46** |
| Attention Masking, $\rho_{\text{th}} = 0.99$ | 88.70 | 2.52 | 2.51 | 86.08 | 65.07 | 64.06 | 42.02 | 77.04 |
| BDHS$_{\text{attn}}$, $\rho_{\text{th}} = 0.98$ | **88.80** | 2.56 | 2.68 | **86.54** | 65.02 | 64.03 | 43.03 | 76.10 |
| BDHS$_{\text{attn}}$, $\rho_{\text{th}} = 0.99$ | 88.75 | 2.61 | 2.71 | 86.33 | 65.07 | 63.97 | 43.39 | 75.58 |
| BDHS$_{\text{attn}}$, $\rho_{\text{th}} = 1.00$ | 88.70 | 2.63 | 2.80 | 84.15 | **65.18** | 63.93 | 43.12 | 75.37 |
| BDHS$_{\text{noise}}$, $N = 100$ | 88.50 | 2.58 | 2.48 | 82.46 | 64.96 | **64.34** | 40.14 | 75.47 |
| BDHS$_{\text{noise}}$, $N = 200$ | 88.55 | 2.49 | 2.38 | 83.43 | 65.10 | 64.24 | 38.76 | 74.53 |
| BDHS$_{\text{noise}}$, $N = 300$ | 88.59 | 2.43 | 2.45 | 85.16 | 65.11 | 64.18 | 40.69 | 76.10 |
| BDHS$_{\text{noise}}$, $N = 400$ | 88.66 | 2.39 | 2.42 | 83.72 | 65.09 | 64.29 | 40.41 | 75.16 |
| BDHS$_{\text{noise}}$, $N = 500$ | 88.59 | 2.36 | 2.49 | 84.53 | 65.05 | 64.14 | 41.38 | 75.16 |

Table 15: Additional ablation results for Offline-BDHS. All results are based on LLaVA 1.6-7B, using DPO and the POVID (5k) sample for the source of images and prompts. and prompt.

For the majority of benchmarks, the variants with BDHS non-preferred responses improve over the non-BDHS datasets, especially for LLaVA$^W$ and MMVet.

## G.7 ADDITIONAL EXAMPLES

This section presents further examples of responses generated from LLaVA instruct prompts and images. The different variants of BDHS are introduced in Section 3.1.

Figure 9 presents generated responses for a selected example defined by image, prompt and SFT ground truth from the LLaVA Instruct dataset. This example should particularly demonstrate the difference between attention masking and noisy images. BDHS with attention masking ($N_{\text{BDHS}} = 5$) is referred to as as BDHS$_{\text{attn}}$ and BDHS with noisy images in the input as BDHS$_{\text{noise}}$. For $\rho_{\text{th}} = 0$ attention masking is disabled but still guided along the ground truth response. The model is able to properly identify the parking meter in the image. With increased attention masking the model starts to hallucinate as desired. Even with fully masked image embeddings the model still hallucinates, while for BDHS$_{\text{noise}}$ the generated responses tend to refer to the blurriness of the images. The example includes responses for the teacher-forced POVID-style image distortion as described in Section G.1.

| Dataset | POPE ↑ | MMHAL ↑ | MMHAL$^V$ ↑ | LLaVA$^W$ ↑ | VQA$^T$ ↑ | GQA ↑ | MMVet ↑ | Recall$^{coco}$ ↑ |
|---|---|---|---|---|---|---|---|---|
| POVID (17k) | 88.09 | 3.16 | 3.07 | 78.63 | 64.56 | 64.12 | 40.60 | 73.48 |
| BDHS (POVID, 17k) | 89.09 | 2.90 | 2.91 | 80.49 | 65.26 | 64.34 | 43.35 | 71.07 |
| VLFeedback (GPT-4V resp., 34k) | 86.59 | 3.05 | 2.93 | 82.91 | 65.02 | 63.86 | 37.62 | 69.60 |
| BDHS (VLFeedback, GPT-4V resp., 34k) | 86.72 | 3.10 | 2.73 | 88.82 | 65.23 | 63.87 | 42.16 | 72.11 |

Table 16: Additional results for BDHS with $\rho_{th} = 0.99$ (DPO) for larger datasets, i.e. full POVID (17k) and a 34k VLFeedback variant with GPT-4V responses as described in Section 4.3.

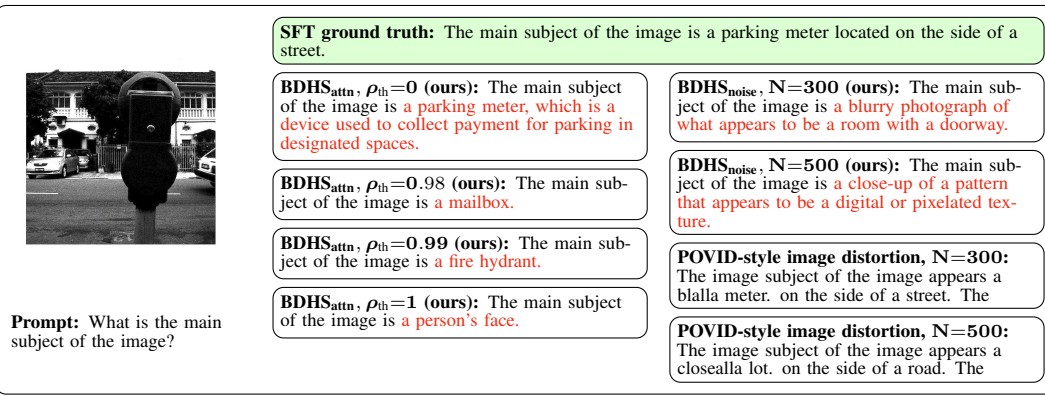

Figure 9: Example of generated responses for different hyperparameters and approaches. The image, prompt and SFT ground truth are taken from LLaVA Instruct. For guided generation, actual model completions are shown in bold face.

Due to the token-based, teacher-forced predictions, the generated responses often are non-sensical and inconsistent which worsens for higher noise levels.

Further examples are shown in Figure 10.

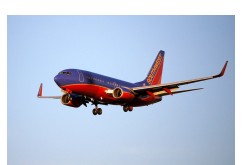

**Prompt:** What color is the jetliner in the image?

> **SFT ground truth:** The jetliner in the image is blue and red.

| | |
|---|---|
| **BDHS**$_{\text{attn}}$**, $\rho_{\text{th}}$=0 (ours):** The jetliner in the image is red and blue. | **BDHS**$_{\text{noise}}$**, N=300 (ours):** The jetliner in the image is blue. |
| **BDHS**$_{\text{attn}}$**, $\rho_{\text{th}}$=0.98 (ours):** The jetliner in the image is white. | **BDHS**$_{\text{noise}}$**, N=500 (ours):** The jetliner in the image is not visible due to the high-resolution pixelation. |
| **BDHS**$_{\text{attn}}$**, $\rho_{\text{th}}$=0.99 (ours):** The jetliner in the image is white. | **POVID-style image distortion, N=300:** The jetliner in the image is blue. red. |
| **BDHS**$_{\text{attn}}$**, $\rho_{\text{th}}$=1 (ours):** The jetliner in the image is white. | **POVID-style image distortion, N=500:** The imageliner in the image is not. white. |

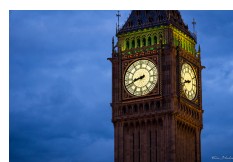

**Prompt:** What is the subject of the image?

> **SFT ground truth:** The subject of the image is the Big Ben clock tower, which is a famous attraction in Europe, particularly in London, England.

| | |
|---|---|
| **BDHS**$_{\text{attn}}$**, $\rho_{\text{th}}$=0 (ours):** The subject of the image is the Elizabeth Tower, more commonly known as Big Ben, a famous clock tower located in London, England. | **BDHS**$_{\text{noise}}$**, N=300 (ours):** The subject of the image is the Elizabeth Tower, which is the clock tower at the north end of the Palace of Westminster in London, England. |
| **BDHS**$_{\text{attn}}$**, $\rho_{\text{th}}$=0.98 (ours):** The subject of the image is a clock tower with a blue sky background. | **BDHS**$_{\text{noise}}$**, N=500 (ours):** The subject of the image is a close-up of a pattern that appears to be a digital or pixelated texture. |
| **BDHS**$_{\text{attn}}$**, $\rho_{\text{th}}$=0.99 (ours):** The subject of the image is a tall, narrow tower with a pointed roof, which appears to be a church steeple. | **POVID-style image distortion, N=300:** The subject of the image is the Elizabeth Ben clock tower in which is a famous landraction in London, specifically in London, England. The |
| **BDHS**$_{\text{attn}}$**, $\rho_{\text{th}}$=1 (ours):** The subject of the image is a person's hand holding a smartphone. | **POVID-style image distortion, N=500:** The image of the image appears a pattern Bang clock tower in which is a famous landraction in London, specifically in London, England. The |

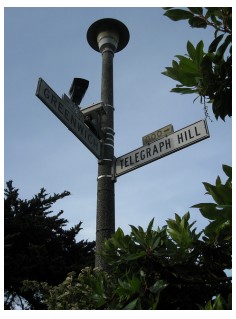

**Prompt:** What street names are visible on the street sign in the image?

> **SFT ground truth:** The street names visible on the street sign in the image are "Telegraph Hill" and "Greenwich Road."

| | |
|---|---|
| **BDHS**$_{\text{attn}}$**, $\rho_{\text{th}}$=0 (ours):** The street sign in the image shows "Greenwich" and "Telephone Hill. | **BDHS**$_{\text{noise}}$**, N=300 (ours):** The image is quite blurry, but I can make out the street sign. |
| **BDHS**$_{\text{attn}}$**, $\rho_{\text{th}}$=0.98 (ours):** The street sign in the image shows the intersection of "Telephone Pole" and "Newspaper Box" streets. | **BDHS**$_{\text{noise}}$**, N=500 (ours):** The image you've provided is very blurry and lacks clear details, making it difficult to read any street names. |
| **BDHS**$_{\text{attn}}$**, $\rho_{\text{th}}$=0.99 (ours):** The street sign in the image shows the intersection of "Broadway" and "Spring Street. | **POVID-style image distortion, N=300:** The image sign visible on the street sign in the image are "Hchn"" and "H Street Street." |
| **BDHS**$_{\text{attn}}$**, $\rho_{\text{th}}$=1 (ours):** The street sign in the image shows the intersection of "Main Street" and "Elm Street. | **POVID-style image distortion, N=500:** The image sign are on the street sign in the image are "1chn Avenue" and "Bwich Street." |

Figure 10: Examples of generated responses from BHDS ablations and POVID-style image distortion. The image, prompt and SFT ground truth are taken from LLaVA-Instruct-150k, which sources them from CoCo (Lin et al., 2014).

