# OpenReview forum: "Understanding Alignment in Multimodal LLMs: A Comprehensive Study"
_ICLR.cc/2025/Conference — Submitted to ICLR 2025_

### Official Review · Reviewer_Z5VU · 2024-11-01

**Soundness:** 3
**Presentation:** 3
**Contribution:** 2
**Rating:** 6
**Confidence:** 3

**Summary:**

The paper explores preference alignment for improving Multimodal Large Language Models (MLLMs), specifically focusing on reducing hallucinations and increasing alignment between model outputs and image content. It provides a thorough analysis of various alignment methods and introduces a novel approach, Bias-Driven Hallucination Sampling (BDHS), which effectively generates preference data without human annotation or external models.

**Strengths:**

1. The study systematically compares offline and online alignment methods, examining their impact on model performance across various metrics like hallucination reduction and response quality.
2. BDHS presents a low-cost, innovative solution to generate preference data, showing competitive results against other data-heavy methods.

**Weaknesses:**

1. While the paper examines alignment techniques and datasets, it does not clearly articulate the primary findings from these investigations, which can make it challenging for readers to grasp the significance and implications of the study

2. BDHS demonstrates promising results; however, its effectiveness may differ across various MLLMs and visual tasks. Conducting additional experiments with diverse model architectures would bolster claims regarding its generalizability.

**Questions:**

1. What are the effects of scaling up BDHS in terms of data size or complexity on model performance?
2. What specific modifications could be made to BDHS to achieve state-of-the-art results on key benchmarks?

---

> ### Author Response · Authors · 2024-11-19
> **Official Comment by Authors**
>
> Thank you so much for your feedback. Please find our answers to your questions below.
>
> **Q1: What are the effects of scaling up BDHS in terms of data size or complexity on model performance?**
>
> * BDHS is used to generate the “rejected” responses, and therefore naturally scales with the number and distribution of ground truth/preferred responses. If the reference dataset grows with a limited increase in diversity, the generated BDHS responses could eventually theoretically become indistinguishable from the preferred ones (no inherent hallucinations triggered). Our algorithm would then discard any further growth of the dataset by evaluating the similarities through sequence embeddings. However, in our experiments with common alignment datasets, we did not observe saturation. To support this with data, we have added a new table (Table 16) to our appendix that compares larger BDHS dataset variants with their non-BDHS references, specifically for the full POVID dataset (17k samples) and our custom VLFeedback dataset with GPT-4V preferred responses (34k samples). In both cases, the BDHS variants outperform the non-BDHS datasets for the majority of benchmarks, particularly for LLaVA in the wild and MMVet, similar to the smaller 5k ablations we showed before.
>
> **Q2: What specific modifications could be made to BDHS to achieve state-of-the-art results on key benchmarks?**
>
> * The goal of our alignment approach is to reduce hallucinations rather than inject additional knowledge into the model. As a result, the observed improvements primarily reflect reductions in hallucinations rather than enhancements in domain-specific capabilities (e.g., mathematics). This focus is why the most significant gains are seen on hallucination benchmarks. For instance, on the POPE benchmark, our aligned model achieves SOTA performance and is highly competitive on Objhal-v, even against models trained on data generated by stronger sources like GPT-4V or human annotators.
> To further enhance the BDHS pipeline, one promising direction could involve integrating our method with ranking-based approaches, such as the online-DPO pipeline. In this framework, instead of deriving both preference pairs from the model in every iteration, we could use BDHS to generate samples for a portion of the training data. This hybrid approach could leverage the strengths of both methods, potentially driving further improvements across various benchmarks. We view this as an exciting avenue for future research.

---

### Official Review · Reviewer_JkXP · 2024-11-04

**Soundness:** 3
**Presentation:** 3
**Contribution:** 2
**Rating:** 6
**Confidence:** 3

**Summary:**

This paper investigates preference alignment techniques for Multimodal Large Language Models (MLLMs), focusing on how they address hallucinations, which occur when models produce responses not grounded in visual inputs. The study categorizes alignment methods into offline and online approaches and examines various multimodal preference datasets. The authors propose a novel data generation method called Bias-Driven Hallucination Sampling (BDHS), which does not require human annotations or external models. Experimental results demonstrate BDHS’s effectiveness compared to more resource-intensive methods.

**Strengths:**

1、Comprehensive Analysis: The paper provides a detailed comparison of alignment methods, including offline and online strategies, and evaluates their effectiveness using diverse datasets.


2、Novel Data Generation Method: The introduction of BDHS offers a cost-effective alternative to traditional alignment approaches, reducing the need for human annotation or external supervision while maintaining competitive performance.

**Weaknesses:**

1、Clarification of Methodological Choices: It would be helpful to better understand why specific thresholds and parameters were chosen for BDHS, such as the similarity score threshold and masking strategy.


2、Generalizability of BDHS: It remains unclear whether BDHS can be effectively applied to models beyond the specific ones studied. Further discussion on its applicability to other MLLMs or domains would strengthen the paper.

**Questions:**

1、The DHS method induces hallucinations by performing attentional masking in the latent space. Is it possible that this strategy could affect the sensitivity of the model to critical details in the image? Have ablation experiments been performed to quantify the effect of this attentional masking in scenes of varying visual complexity? In addition, how to select the range of attention masking, and whether the alignment effect can be optimized by dynamic adjustment?

2、The paper mentions filtering out different non-preferred responses by semantic similarity score. For this filtering mechanism, is it possible that there is a bias that makes the model perform better or worse on specific types of semantic content? Have comparative experiments with different similarity scoring models been conducted to confirm the robustness of the selection mechanism? Furthermore, could this similarity score lead to a tendency for models to oversimplify when faced with less common or more complex visual scenes?

3、Does the performance of the BDHS method on the LLaVA 1.6-7B model generalize to larger or smaller model sizes? Have any experiments been conducted on models with different parameter numbers to explore whether this approach exhibits different advantages or disadvantages depending on the model size? Especially on small-scale models, is it possible that the method effect is not significant due to parameter limitations?

4、To what extent do current hallucina-evaluation benchmarks such as POPE and MMHALBench-V truly reflect model performance in real-world applications?

---

> ### Author Response · Authors · 2024-11-19
> **Official Comment by Authors**
>
> We are very grateful for your detailed review and feedback. Please find our response to your questions below.
>
>
> **Q1: ... How to select the range of attention masking, and whether the alignment effect can be optimized by dynamic adjustment?**
>
> - Since we use BDHS to generate “rejected” responses, our primary goal was to ensure that these responses are semantically different from the “chosen” responses. To achieve this, we designed an iterative approach where a rejected response is added to our dataset only if it meets a specific threshold on the similarity metric (a hyperparameter determined through our experiments), ensuring it is not too similar to the chosen response. In this process, as you may have suspected, we observed that simple question-answer pairs typically require more iterations to produce a response that is sufficiently distinct from the chosen response. We conducted extensive ablation studies on the amount of masking (Appendix G) and compared the sensitivity of our approach to that of other state-of-the-art methods designed for similar tasks (POVID dataset). Our results demonstrate that our approach is both more effective and less sensitive to the hyperparameter. That said, exploring how the amount of masking required to (more) reliably introduce hallucination correlates with visual complexity is an interesting idea, and we plan to explore its impact in future research.
>
> **Q2: ... Could this similarity score lead to a tendency for models to oversimplify when faced with less common or more complex visual scenes?**
>
> - Our approach relies on the ability to measure the similarity between the embeddings of chosen and rejected responses. Most recent sentence embeddings perform well in this task, making our pipeline relatively robust to the specific choice of embedding. After evaluating multiple options, we empirically found all-mpnet-base-v2 to be an effective and computationally efficient choice. This computational efficiency is particularly important in an online setting. Furthermore, by employing SFT forcing, we avoid oversimplifying the rejected responses. In most cases, the model generates only part of the “rejected” response rather than rewriting the entire sentence. This is a crucial aspect of the BDHS algorithm, which ensures that the rejected responses remain minimally different from the chosen responses in terms of complexity and style, while being semantically distinct. For example, two responses might describe the color of an object using the same style, but one specifies the correct color while the other specifies an incorrect one. This design ensures that the model must comprehend the image in order to optimize the objective effectively.
>
> **Q3: ... Especially on small-scale models, is it possible that the method effect is not significant due to parameter limitations?**
>
> - We have not conducted experiments on smaller models. Initially, we started our experiments on weaker models, specifically the LLaVa 1.5 family. However, we decided against continuing with them because, although they were widely used by the community at the time, we found that weaker models had such a large margin for improvement that almost any intervention proved effective. For example, higher values of 𝛽 in the DPO setting consistently resulted in better performance. This suggests that diverging from the reference policy enhances the model’s capabilities, which may indicate underlying weaknesses in the base model. Instead, we focused our study on stronger models after the SFT stage to better highlight the benefits that the alignment stage can bring to multimodal LLMs. While we conducted some limited experiments on larger models (e.g., LLaVa 1.6 13b), these followed similar patterns to the 7b model, however were not comprehensive enough to include in the paper due to compute limitations. Based on our findings in this paper and our ongoing research in this area, we believe this approach is consistently effective in settings where we can observe LLM bias in multimodal LLM behavior.
>
> **Q4: To what extent do current hallucination evaluation benchmarks such as POPE and MMHALBench-V truly reflect model performance in real-world applications?**
>
> * This is certainly one of the areas our research identified as having significant room for exploration, which we discussed in Appendix B2-B4. We found our proposed MMHALBench-V to be an effective benchmark for evaluating hallucination; however, we acknowledge that there is still plenty of scope for further improvement. For the experiments presented in the paper, we carefully analyzed both the wins and losses associated with any observed advancements to ensure they genuinely reflected enhanced model performance. For example, although we observed substantial gains in results on ObjhalBench, we excluded those numbers due to the issues outlined in Appendix B3.

---

### Official Review · Reviewer_a6hU · 2024-11-08

**Soundness:** 3
**Presentation:** 3
**Contribution:** 3
**Rating:** 8
**Confidence:** 3

**Summary:**

The paper addresses challenges in aligning MLLMs with human preferences to improve response accuracy and reduce hallucinations. It reviews various offline and online alignment strategies, including DPO  and RLHF, and introduces BDHS. BDHS generates preference data without human annotation, leveraging model-inherent biases to enhance performance cost-effectively. Results indicate BDHS is competitive with established preference datasets, demonstrating its potential as a lightweight alternative to traditional alignment approaches for MLLMs, especially in tasks requiring high fidelity between visual inputs and textual responses.

**Strengths:**

1. The paper introduces a unique approach to generate preference data for MLLMs by utilizing model biases without human or external model annotations.
2. The paper provides empirical analysis, comparing BDHS with other alignment methods across multiple benchmarks, highlighting its effectiveness and resource efficiency in aligning MLLMs.

**Weaknesses:**

1. The proposed data sampling approach partially mitigates hallucination issues in MLLMs but does not completely resolve them.
2. The BDHS method's dependency on hyperparameters, such as mask thresholds, could affect reproducibility across different model implementations.

**Questions:**

1. Will the code be open-sourced to facilitate further research in this area?
2. How does the proposed approach ensure that the distribution of generated hallucination data aligns with real-world hallucination data distributions?

---

> ### Author Response · Authors · 2024-11-19
> **Official Comment by Authors**
>
> Thank you so much for your positive feedback and recognizing the contribution of our work. Please find the answer to your questions below:
>
>
> **Q1: Will the code be open-sourced to facilitate further research in this area?**
>
>
> - Yes, we are working on open sourcing the codebase.
>
>
> **Q2: How does the proposed approach ensure that the distribution of generated hallucination data aligns with real-world hallucination data distributions?**
>
>
> - Hallucinations in multimodal LLMs can arise from various root causes, with bias in the language model being a significant contributor. Our proposed approach, BDHS, is specifically designed to address this bias directly. While BDHS does not explicitly target other causes of hallucinations, such as insufficient real-world knowledge, we have observed overall improvements across multiple benchmarks, indicating a positive impact on reducing hallucinations. We have added some explanation to the conclusion to highlight your feedback on this area further.
> To further ensure that the distribution of generated hallucination data aligns closely with real-world distributions, we introduced an online version of BDHS. This online approach allows the model to dynamically adjust the hallucinations as it learns, ensuring that as the model's performance improves, the generated hallucinations are continually aligned with the improved model.

---

> > ### Comment · Reviewer_a6hU · 2024-11-27
> >
> > Thnaks for the response! My concerns have been addressed, so I will retain my rating of “8: Accept, good paper.”

---

### Author Response · Authors · 2024-11-19
**General response to the reviewers**

We sincerely thank all the reviewers for their time and effort to review our work. We are also grateful to our reviewers for recognizing the novelty and contribution of our research and providing thoughtful feedback. We have updated the submitted PDF to address the feedback and comments of the reviewers. Please find our responses to the questions and clarifications of key points below.

---

### Meta-Review · Area_Chair_Trie · 2024-12-22

**Metareview:**

Paper analyzes off-line and on-line MLLM / VLM human preference alignment approaches, including DPO and PPO and introduces its own variant -- BDHS. BDHS generates preference data automatically, by leveraging VQA pairs and generating negative pairings through LLM inherent biases using masking.

The paper was reviewed by three expert reviewers and received: 1 x accept, good paper and 2 x marginally above the acceptance threshold ratings. Overall reviewers agree that the approach is sound and presentation is good, but differ in the view of overarching contributions (with two reviewers rating it as "Fair"). Additional concerns with the work revolved around a few core issues: (1) proposed data sampling approach seems to only have limited effectiveness [a6hU], (2) stability with respect to hyper parameters (e.g., mask threshold) [a6hU] and selection [JkXP], (3) limited evaluation of BDHS method, mainly on the LLaVA 1.6-7B [JkXP], and (4) lacking clear articulation of findings which makes it more difficult to ascertain significance of the work [Z5VU]. Authors have provided a rebuttal and while [a6hU] appears to be convinced by the arguments, he/she did not further update the score. Unfortunately, the remaining two reviewers [JkXP] and [Z5VU] did not engage in discussion.

AC has read the reviews, the rebuttal and the discussion; also looked and read the paper itself. With respect to concerns raised by [JkXP] and [Z5VU], AC didn't find arguments from the rebuttal completely convincing. Specifically, it would be good to see that BDHS approach's utility expands beyond LLaVA family of models and models of different sizes. In other words, while the findings are interesting and suggestive, the full impact on the breadth of MLLM / VLM models is not clear.

Given the borderline nature of the paper, it was also discussed at length between AC and Senior AC. Given that none of the reviewers strongly championed the paper, and considering that some concerns still remain, AC and Senior AC, collectively, are recommending Rejection at this time. Authors are encouraged to address the raised concerns and to resubmit the paper to the next venue.

Also, while AC wants to be clear that this has NOT been taken into account in evaluation of the work, given that this is concurrent, authors are encouraged to look at the "Unpacking DPO and PPO: Disentangling Best Practices for Learning from Preference Feedback" paper from NeurIPS 2024, which is closely related in terms of their analysis.

**Additional Comments On Reviewer Discussion:**

Authors have provided a rebuttal and while [a6hU] appears to be convinced by the arguments -- stating "My concerns have been addressed", he/she did not further update the score. The remaining two reviewers [JkXP] and [Z5VU] did not engage in discussion, as such AC did the best in ascertaining how their comments have been addressed by the authors. Overall, the rebuttal was found to be only partially convincing and some concerns remained, leading to the discussion between AC and Senior AC and the assessment and recommendation above.

---

### Decision · Program_Chairs · 2025-01-22

Reject